# Zeroth-Order Sampling Methods for Non-Log-Concave Distributions: Alleviating Metastability by Denoising Diffusion

**Ye He**
Georgia Institute of Technology
yhe367@gatech.edu

**Kevin Rojas**
Georgia Institute of Technology
kevin.rojas@gatech.edu

**Molei Tao**
Georgia Institute of Technology
mtao@gatech.edu

## Abstract

This paper considers the problem of sampling from non-logconcave distribution, based on queries of its unnormalized density. It first describes a framework, Denoising Diffusion Monte Carlo (DDMC), based on the simulation of a denoising diffusion process with its score function approximated by a generic Monte Carlo estimator. DDMC is an oracle-based meta-algorithm, where its oracle is the assumed access to samples that generate a Monte Carlo score estimator. Then we provide an implementation of this oracle, based on rejection sampling, and this turns DDMC into a true algorithm, termed Zeroth-Order Diffusion Monte Carlo (ZOD-MC). We provide convergence analyses by first constructing a general framework, i.e. a performance guarantee for DDMC, with**out** assuming the target distribution to be log-concave or satisfying any isoperimetric inequality. Then we prove that ZOD-MC admits an inverse polynomial dependence on the desired sampling accuracy, albeit still suffering from the curse of dimensionality. Consequently, for low dimensional distributions, ZOD-MC is a very efficient sampler, with performance exceeding latest samplers, including also-denoising-diffusion-based RDMC and RSDMC. Last, we experimentally demonstrate the insensitivity of ZOD-MC to increasingly higher barriers between modes or discontinuity in non-convex potential.

## 1 Introduction

The problem of drawing samples from a distribution based on unnormalized density $\propto \exp(-V)$ (described by the potential $V$) is a fundamental statistical and algorithmic problem. This classical problem nevertheless remains as a research frontier, providing pivotal tools to applications such as decision making, statistical inference / estimation, uncertainty quantification, data assimilation, and molecular dynamics. Worth mentioning is that machine learning could benefit vastly from progress in sampling as well, not only because of its connection to inference, optimization and approximation, but also through modern domains such as diffusion generative modeling & differential privacy.

Recent years have seen rapid developments of sampling algorithms with quantitative and non-asymptotic theoretical guarantees. Many of the results are either based on discretizations of diffusion processes [12, 13, 50, 16, 34, 33] or gradient flows [38, 10, 22]. In order to develop such guarantees, it is necessary to make assumptions about the target distributions, for instance, that it satisfies an isoperimetric property, where standard requirements are log-concavity or functional inequalities

38th Conference on Neural Information Processing Systems (NeurIPS 2024).

[12, 56, 21, 8, 48]. However, there is empirical evidence that the corresponding algorithms struggle to sample from targets that have high barriers between modes that create metastability. Overcoming such issues is highly nontrivial and researchers have continued to develop new methods to tackle these problems.

Diffusion models have lately shown remarkable ability in the generative modeling setting, with applications including image, video, audio, and macromolecule generations. This created a wave of theoretical work that showed the ability of diffusion models to sample from distributions under minimal assumptions [14, 57, 5, 32, 29, 4, 11, 3]. However, these works all started with the assumption that there is access to an approximation of the score function with some accuracy. This is a reasonable assumption for the task of generative modeling when one spends enough efforts on the training of the score, but the task of sampling is different. A natural question is: can we leverage the insensitivity of diffusion models to multimodality to efficiently sample from unnormalized, non-log-concave density? This would require approximating the score, which is then used as an inner loop inside an outer loop that integrates reverse diffusion process to transport, e.g., Gaussian initial condition, to nearly the target distribution.

The seminal works by [26, 27, 20] try to answer this question using Monte Carlo estimators of the score function and to provide theoretical guarantees. We also mention earlier work by [53] which learns parameterized scores and more work along the same line by [58, 44, 54, 55], whose theoretical guarantees are less clear but are based other interesting ideas. [26] proposed Reverse Diffusion Monte Carlo (RDMC), which estimates the score via LMC algorithm and relaxes the isoperimetric assumptions in the analysis of traditional sampling algorithms. [20] proposed a similar method, stochastic localization via iterative posterior sampling (SLIPS), which approximate the score via Metropolis-adjusted Langevin algorithm (MALA). However, both methods rely on the usage of a small time window where isoperimetric properties hold. This leaves the problem of finding a good initialization for the diffusion process. To alleviate this issue, [27] developed an acceleration of RDMC, the Recursive Score Diffusion-based Monte Carlo (RSDMC), which improves the non-asymptotic complexity to be quasi-polynomial in both dimension and inverse accuracy and gets rid of any isoperimetric assumption. Such work provides strong theoretical guarantees, however it requires a lot of computational power to get a high accuracy sampler. Additionally, RDMC, SLIPS and RSDMC are all based on first-order queries (i.e. gradients of $V$), which brings extra computational and memory costs, in addition to requiring a continuous differentiable $V$. Motivated by these two observations, we create a sampler that only makes use of **zeroth-order** queries without assuming any isoperimetric conditions on the target distribution. Our contributions can be summarized as follows.

- We introduce an oracle-based meta-algorithm **DDMC (Denoising Diffusion Monte Carlo)** and provide a non-asymptotic guarantee in KL-divergence in Theorem 1. Our result provides theoretical insight on the choice of optimal step-size in DDMC as well as in denoising diffusion models (Sec. 3.2).

- We develop a novel algorithm **ZOD-MC (Zeroth Order Diffusion-Monte Carlo)** that uses zeroth-order queries and the global minimal value of the potential function to generate samples approximating the target distribution. In Corollary 3.1, we establish a zeroth-order query complexity upper bound for general target distributions satisfying mild smoothness and moment conditions. Our result is summarized and compared to other sampling algorithms in Table 5.

- The advantages of our algorithm are experimentally verified for non-log-concave target distributions. We demonstrate the insensitivity of our algorithm to various high barriers between modes, and the ability of correctly account for discontinuities in the potential.

## 2 Preliminaries

### 2.1 Diffusion Model

Diffusion model generates samples that are similar to training data, by requiring the generated data to follow the same latent distribution $p$ as the training data. To do so, it considers a forward noising

---

[1] Assumption [A4] in [26] is a soft version of strongly log-concave outside a ball.

[2] This criterion measures the KL-divergence from the output distribution to a distribution that is closed to the target distribution in Wasserstein-2 distance. This criterion is considered in analyzing denoising diffusion models with step-size that accommodates the early stopping technique, see [4, 3]. This criterion does not apply to RDMC/RSDMC since they don't use early stopping and instead assume the target distribution to be smooth and fully supported on $\mathbb{R}^d$. See Sec.3.3 for more discussions.

| Algorithms | Queries | Assumptions | Criterion | Oracle Complexity |
|---|---|---|---|---|
| LMC | first-order | LSI | KL | $\mathcal{O}(d\varepsilon^{-1})$ |
| RDMC | first order | soft log-concave[1] | TV | $\exp(\mathcal{O}(\log(d))\tilde{\mathcal{O}}(\varepsilon^{-1}))$ |
| RSDMC | first-order | None | KL | $\exp(\mathcal{O}(\log^3(d\varepsilon^{-1})))$ |
| Proximal Sampler | zeroth-order | log-concave | KL | $\mathcal{O}(d\varepsilon^{-1})$ |
| ZOD-MC | zeroth-order | None | KL $+ W_2$[2] | $\exp(\tilde{\mathcal{O}}(d)\mathcal{O}(\log(\varepsilon^{-1})))$ |

Table 1: Comparison of ZOD-MC to LMC, RDMC, RSDMC and the Proximal Sampler: Summary of isoperimetric assumptions and oracle complexities to generate a $\varepsilon$-accurate sample under different criterion. $\tilde{O}$ hides polylog($d\varepsilon^{-1}$) factors. The zeroth-order oracle complexity of ZOD-MC is from Corollary 3.1 for achieving both $\varepsilon$ KL and $\varepsilon$ $W_2$ errors. These theoretical results suggest that in the absense of isoperimetric assumptions, ZOD-MC excels in low-dimensions.

process that transforms a random variable into Gaussian noise. One most commonly used forward process is (a time reparameterization of) the Ornstein-Uhlenbeck (OU) process, given by the SDE:

$$\mathrm{d}X_t = -X_t\mathrm{d}t + \sqrt{2}\mathrm{d}B_t, \quad X_0 \sim p, \tag{1}$$

where $\{B_t\}_{t\geq 0}$ is the standard Brownian motion in $\mathbb{R}^d$. The OU process that solves (1) is in distribution equivalent to a sum of two independent random vectors: $X_t = e^{-t}X_0 + \sqrt{1 - e^{-2t}}Z$ where $(X_0, Z) \sim p \otimes \gamma^d$ and $\gamma^d$ is the standard Gaussian distribution in $\mathbb{R}^d$. Denote $p_t = \mathrm{Law}(X_t)$ for all $t \geq 0$. If we consider a large, fixed terminal time $T$ of (1), then $p_T$ is close to $\gamma^d$. Then, the denoising or backwards diffusion process, $\{\bar{X}_t\}_{0\leq 0\leq T}$, can be constructed by reversing the OU process from time $T$, meaning that $\mathrm{Law}(\bar{X}_t) := \mathrm{Law}(X_{T-t})$ for all $t \in [0, T]$. By doing so we obtain the denoising diffusion process which solves the following SDE:

$$\mathrm{d}\bar{X}_t = (\bar{X}_t + 2\nabla \log p_{T-t}(\bar{X}_t))\mathrm{d}t + \sqrt{2}\mathrm{d}\bar{B}_t, \quad \bar{X}_0 \sim p_T,\ 0 \leq t \leq T, \tag{2}$$

where $\{\bar{B}_t\}_{0\leq t\leq T}$ is a Brownian motion in $\mathbb{R}^d$, independent of $\{B_t\}_{0\leq t\leq T}$ and $\nabla \ln p_t$ is usually referred as the score function for $p_t$. Although the denoising process initializes at $p_T$, we can't generate exact samples from $p_T$. In practice, people consider the standard Gaussian initialization $\gamma^d$ due to the fact that $p_T$ is close to $\gamma^d$ when $T$ is large. The denosing process with the standard Gaussian initialization is given by

$$\mathrm{d}\tilde{X}_t = (\tilde{X}_t + 2\nabla \log p_{T-t}(\tilde{X}_t))\mathrm{d}t + \sqrt{2}\mathrm{d}\bar{B}_t, \quad \tilde{X}_0 \sim \gamma^d,\ 0 \leq t \leq T. \tag{3}$$

By simulating this denoising process (3), we can achieve the goal of generating new samples. However, the denoising process (3) can't be simulated directly due to the fact that the score function is not explicitly known. A widely applied method to solve this issue is to learn the score function through denoising score matching [51, 24, 52]. Given a learned score, denoted as $s(t, x)$, one can simulate the denoising diffusion process using discretizations like the Euler Maruyama or some exponential integrator. From a theoretical perspective, assuming the learned score satisfies

$$\mathbb{E}_{x\sim p_t}\left[\,\|s(t, x) - \nabla \ln p_t(x)\|^2\,\right] \leq \epsilon_{\mathrm{score}}^2, \quad \forall\, 0 \leq t \leq T, \tag{4}$$

non-asymptotic convergence guarantees for diffusion models are obtained in [5, 3, 4, 11]. For instance, in [3], polynomial iteration complexities were proved without assuming any isoperimetric property of the data distribution and only assuming the data distribution has a finite second moment and a score estimator satisfying (4) is available.

In this work, we consider instead the sampling setting, in which no existing samples from the target distribution is available. Our sampling algorithm and theoretical analysis are motivated from the denoising diffusion process given by (2) and its corresponding discretization through the exponential integrator in Algorithm 1. In particular, we first introduce an oracle-based meta-algorithm, DDMC, which integrates Algorithm 1 and Algorithm 2, where the exponential integrator scheme of (2) is applied to generate samples and the score function is approximated by a Monte Carlo estimator assuming independent samples from a conditional distribution are available.

## 2.2  Rejection Sampling and Restricted Gaussian Oracle

Rejection sampling is a popular Monte Carlo method for sampling a target distribution, $p$, based on the zeroth-order queries of the potential $V$. It requires that we have access to the potential function

---

**Algorithm 1:** Denoising Diffusion Sampling via Exponential Integrator

---

**Input** : $N \in \mathbb{Z}_+, 0 = t_0 < \cdots < t_N = T - \delta$, score estimator $\{s(T - t_k, \cdot)\}_{k=0}^{N-1}$.
**Output** : $x_N$.
generate a sample $x_0 \sim \gamma^d$;
**for** $k = 0, 1, \cdots, N - 1$ **do**
$\quad$ generate $\xi_k \sim \gamma^d$ such that $\xi_k$ is independent to $\xi_0, \cdots, \xi_{k-1}$;
$\quad$ $x_{k+1} \leftarrow e^{t_{k+1}-t_k} x_k + 2(e^{t_{k+1}-t_k} - 1)s(T - t_k, x_k) + \sqrt{e^{2(t_{k+1}-t_k)} - 1}\xi_k.$
**end**

---

$V_\mu$ of some other distribution $\mu$, such that $\mu$ is easy to sample from and $\exp(-V) \leq \exp(-V_\mu)$ globally. Such a distribution $\mu$ is typically called an envelope for the distribution $p$. With an envelope $\nu$, rejection sampling generates samples from $p$ by running the following algorithm till acceptance:

1. Sample $X \sim \mu$,

2. Accept $X$ with probability $\exp(-V(X) + V_\mu(X))$.

The rejection sampling is considered as a high-accuracy algorithm as it outputs a unbiased sample from the target distribution. However, despite such a remarkable property, it has drawbacks. First, it is a nontrivial task to find an envelope for a general target distribution. Second, rejection sampling usually suffers from "curse of dimensionality". Even for strongly logconcave target distributions, the complexity of the rejection sampling increases exponentially fast with the dimension: in expectation it requires $\kappa^{d/2}$ many rejections before one acceptance, where $\kappa$ is the condition number for the potential $V$, see [9].

The Restricted Gaussian Oracle (RGO), which was first introduced in [31], assumes that an accurate sample from distribution $\pi(\cdot|y) \propto \exp\left(-V(\cdot) - \frac{1}{2\eta}\|\cdot - y\|^2\right)$ can be generated for any $y \in \mathbb{R}^d$, $\eta > 0$ and any potential $V$. Implementing the RGO is challenging. It is usually done by rejection sampling. However, most proposed methods [36, 18], are only suitable for small $\eta$.

Our proposed sampling algorithm, ZOD-MC applies the rejection sampling (Algorithm 3) to implement the RGO with a large value of $\eta$. Details on ZOD-MC are introduced in Section 3.1.

## 3 Denoising Diffusion Monte Carlo Sampling

In this section, we first introduce DDMC and ZOD-MC in Section 3.1. Then we provide a convergence guarantee for DDMC in Section 3.2. Last, in Section 3.3, we establish the zeroth-order query complexity of ZOD-MC. Note DDMC is a meta-algorithm that still requires an implementation of its oracle, and ZOD-MC is an actual algorithm that contains such an implementation. The theoretical guarantee of ZOD-MC (Sec. 3.3), therefore, is based on the analysis framework of DDMC (Sec. 3.2).

### 3.1 Denoising Diffusion Monte Carlo and Zeroth-Order Diffusion Monte Carlo

***Denoising Diffusion Monte Carlo (DDMC).*** Let's start with a known but helpful lemma on score representation, derivable from Tweedie's formula [45].

**Lemma 1.** *Let $\{X_t\}_{t\geq 0}$ be the solution to the OU process (1) and $p_t = Law(X_t)$. Then for all $t > 0$,*

$$\nabla \ln p_t(x) = \mathbb{E}_{x_0 \sim p_{0|t}(\cdot|x)}\left[\frac{e^{-t}x_0 - x}{1 - e^{-2t}}\right], \tag{5}$$

*where $p_{0|t}(\cdot|x) \propto \exp\left(-V(\cdot) - \frac{1}{2}\frac{\|\cdot - e^t x\|^2}{e^{2t}-1}\right)$ is the distribution of $X_0$ conditioned on $\{X_t = x\}$.*

This lemma was for example applied in [26] to do sampling based on the denoising diffusion process in (2). For the sake of completeness, we include its proof in Appendix C.5.

Due to (5), to approximate the score function $\nabla \ln p_t(x)$, it suffices to generate samples that approximate $p_{0|t}(\cdot|x)$. [26, 27] proposed to use Langevin-based algorithms to sample from $p_{0|t}(\cdot|x)$. The first step of our work is to generalize this, with refined and more general theoretical analysis later on, by considering an oracle algorithm, DDMC, which assumes independent samples $\{z_{t,i}\}_{i=1}^{n(t)}$ that

---

**Algorithm 2:** Monte Carlo Score Estimation

---

**Input** : $t \in (0, T]$, $x \in \mathbb{R}^d$, $n(t) \in \mathbb{Z}_+$, $\delta(t) > 0$.
**Output** : $s(t, x)$.
**Oracle** : generate independent $\{z_{t,i}\}_{i=1}^{n(t)}$ such that $W_2(\text{Law}(z_{t,i}), p_{0|t}(\cdot|x)) \leq \delta(t)$.
$s(t, x) \leftarrow \frac{1}{n(t)} \sum_{i=1}^{n(t)} \frac{x - e^{-t} z_{t,i}}{1 - e^{-2t}}$.

---

approximate $p_{0|t}(\cdot|x)$ are available. The Monte Carlo score estimator in Algorithm 2 is given by

$$s(t, x) = \frac{1}{n(t)} \sum_{i=1}^{n(t)} \frac{e^{-t} z_{t,i} - x}{1 - e^{-2t}}, \tag{6}$$

where $n(t)$ is the number of samples and $\delta(t)$ is such that $W_2(\text{Law}(z_{t,i}), p_{0|t}(\cdot|x)) \leq \delta(t)$ for all $i$. In Section 3.2, we will discuss how the performance of sampling depends on $n(t), \delta(t)$.

***Zeroth-Order Diffusion Monte Carlo (ZOD-MC).*** Noticing that in Lemma 1, the conditional distribution has a structured potential function: a summation of the target potential and a quadratic function. Therefore, implementing the oracle in DDMC is equivalent to implementing RGO with $y = e^t x$ and $\eta = e^{2t} - 1$. Based on this, we propose ZOD-MC, a novel methodology based on rejection sampling and DDMC. Rejection samplng (Algorithm 3) can generate i.i.d. Monte Carlo samples required in Algorithm 2. Therefore, ZOD-MC, as a combination of rejection sampling and DDMC, can efficiently sample from non-logconcave distributions. See Appendix B for more details.

---

**Algorithm 3:** Rejection Sampling: generating $\{z_{t,i}\}_{i=1}^{n(t)}$ in Algorithm 2

---

**Input** : $x \in \mathbb{R}^d$, zeroth-order queries of $V$.
**Output** : $z$.
**while** *TRUE* **do**
    Generate $(\xi, u) \sim \gamma^d \otimes U[0, 1]$;
    $z \leftarrow e^t x + \sqrt{e^{2t} - 1}\xi$;
    **return** $z$ **if** $u \leq \exp(-V(z) + V^*)$;
**end**

---

**Remark 1.** *(Remark on the optimization step) In theory, we assume an oracle access to the minimum value of $V$. However, in practice we use Newton's method to find a local minimum. Throughout the sampling process we update the local minimum as we explore the search space.*

**Remark 2.** *(Parallelization) Notice that Algorithm 3 can be run in parallel to generate all the $n(t)$ samples required to compute the score. Contrary to methods like LMC that have a sequential nature, this allows our method to be more computationally efficient and reduce the running time. This is a feature that RDMC or RSDMC doesn't benefit as much from.*

### 3.2 Convergence of DDMC

Our oracle-based meta-algorithm, DDMC, provides a framework for designing and analyzing sampling algorithms that integrate the denoising diffusion model and the Monte Carlo score estimation. In this section, we first present an error analysis to the Monte Carlo score estimation in Proposition 3.1, whose proof is in Appendix C.3. After that, we leverage our result in Proposition 3.1 and provide a non-asymptotic convergence result for DDMC in Theorem 1, whose proof is in Appendix C.4.

**Proposition 3.1.** *Let $\{X_t\}_{t \geq 0}$ be the solution of the OU process (1) and $p_t = \text{Law}(X_t)$ for all $t > 0$. If we define $s(t, x) = \frac{1}{n(t)} \sum_{i=1}^{n(t)} \frac{e^{-t} z_{t,i} - x}{1 - e^{-2t}}$ with $\{z_{t,i}\}_{i=1}^{n(t)}$ being a sequence of independent random vectors such that $W_2(\text{Law}(z_{t,1}), p_{0|t}(\cdot|x)) \leq \delta(t)$ for all $t > 0$ and $x \in \mathbb{R}^d$, then we have*

$$\mathbb{E}\big[\|\nabla \ln p_t(X_t) - s(t, X_t)\|^2\big] \leq \frac{e^{-2t}}{(1 - e^{-2t})^2} \delta(t)^2 + \frac{1}{n(t)} \frac{e^{-2t}}{(1 - e^{-2t})^2} \text{Cov}_p(x). \tag{7}$$

***Choice of $\delta(t)$ and $n(t)$.*** The error bound in (7) helps choose the accuracy threshold $\delta(t)$ and the number of samples $n(t)$ to control the score estimation error over different time. In fact, when

$t$ increases, it requires less samples and allows larger sample errors to get a good Monte Carlo score estimator. If we assume $\mathrm{Cov}_p(x) = \mathcal{O}(d)$ for simplicity, then when $t$ is small, the factor $\frac{e^{-2t}}{(1-e^{-2t})^2} = \mathcal{O}(t^{-2})$ and the choice of $\delta(t) = \mathcal{O}(t\varepsilon)$ and $n(t) = \Omega(dt^{-2}\varepsilon^{-2})$ will lead to the $L^2$-error of order $\mathcal{O}(\varepsilon^2)$. When $t$ is large, the factor $\frac{e^{-2t}}{(1-e^{-2t})^2} = \mathcal{O}(e^{-2t})$ and it only requires $\delta(t) = \mathcal{O}(e^t\varepsilon)$ and $n(t) = \Omega(de^{-2t}\varepsilon^{-2})$ to ensure the $L^2$-error is of order $\mathcal{O}(\varepsilon^2)$. In the latter case, the $\delta(t)$ is of a larger order and $n(t)$ is of a smaller order than the first case.

We now analyze the convergence of DDMC. Recall that Algorithm 1 is an exponential integrator discretization scheme of (2) with the time schedule $0 = t_0 < t_1 < \cdots < t_N = T - \delta$ for some $\delta > 0$. In each iteration, $x_{k+1} = e^{t_{k+1}-t_k}x_k + 2(e^{t_{k+1}-t_k}-1)s(T-t_k, x_k) + \sqrt{e^{2(t_{k+1}-t_k)}-1}\xi_k$, where $\xi_k \sim \gamma^d$ and $s(T-t_k, \cdot)$ is the Monte Carlo score estimator generated by Algorithm 2. The trajectory of Algorithm 1 can be piece-wisely characterized by the following SDEs: for all $t \in [t_k, t_{k+1})$,

$$\mathrm{d}\tilde{X}_t = (\tilde{X}_t + 2s(T - t_k, \tilde{X}_{t_k}))\mathrm{d}t + \sqrt{2}\mathrm{d}\tilde{B}_t, \quad \tilde{X}_0 \sim \gamma^d, \ \tilde{X}_{t_k} = x_k. \tag{8}$$

Therefore, the convergence of DDMC is equivalent to the convergence of the process $\{\tilde{X}_t\}_{0 \leq t \leq t_N}$, which could be quantified under mild assumptions on the target distribution. Next, we present the moment assumption on the target distribution and our non-asymptotic convergence theorem.

**Assumption 3.1.** *The distribution $p$ has a finite second moment: $\mathbb{E}_{x\sim p}[\|x\|^2] = \mathrm{m}_2^2 < \infty$.*

**Theorem 1.** *Assume that the target distribution satisfies Assumption 3.1. Let $\{X_t\}_{t\geq 0}$ be the solution of (1) with $p_t := \mathrm{Law}(X_t)$ and $\{\tilde{X}_t\}_{t\geq 0}$ be the solution of (8) with $q_t := \mathrm{Law}(\tilde{X}_t)$. For any $\delta \in (0,1)$ and $T > 1$, let $0 = t_0 < t_1 < \cdots < t_N = T - \delta$ be a time schedule such that $\gamma_k = \Theta(\gamma_{k-1})$ for all $k = 0, 1, \cdots, N-1$, where $\gamma_k := t_{k+1} - t_k$. Then*

$$KL(p_\delta | q_{t_N}) \lesssim \underbrace{(d + \mathrm{m}_2^2)e^{-2T}}_{I} + \underbrace{\sum_{k=0}^{N-1} \frac{\gamma_k e^{-2(T-t_k)}}{(1-e^{-2(T-t_k)})^2}\left(\delta(T-t_k)^2 + \frac{\mathrm{m}_2^2}{n(T-t_k)}\right)}_{II}$$

$$+ \underbrace{\frac{\mathrm{m}_2^2 e^{-2T}\gamma_0}{(1-e^{-2T})^2} + \sum_{k=1}^{N-1} \frac{(d+\mathrm{m}_2^2)\gamma_k\gamma_{k-1}^2}{(1-e^{-2(T-t_k)})(1-e^{-2(T-t_{k-1})})^2} + \sum_{k=0}^{N-1} \frac{d\gamma_k^2}{(1-e^{-2(T-t_k)})^2}}_{III}, \tag{9}$$

*where $\delta(t), n(t)$ are parameters in Algorithm 2.*

**Remark 3.** *In Theorem 1, we characterize $KL(p_\delta | q_{t_N})$ instead of $KL(p | q_{t_N})$ due to the fact that $KL(p | q_{t_N})$ is not well-defined when the target distribution $p$ is not smooth w.r.t. the Lebesgue measure. It turns out that $p_\delta$ is an alternative distribution to look at because $p_\delta$ is smooth for all $\delta > 0$ and $p_\delta$ is close to $p$ when $\delta$ is small (see Proposition C.1). This is a standard treatment, referred to as early stopping, in the score-based generative modeling literature [e.g., 5, 3].*

The terms I, II, III in (9) correspond to the three types of errors in Algorithm 1, the initialization error, the score estimation error and the discretization error, respectively. Such a decomposition is very common in analyses of diffusion models, see [5, 6, 3]. Since we consider the sampling setting, the score estimation error is derived from the error analysis to the Monte Carlo score estimator, i.e., Proposition 3.1. A detailed discussion on these three types of errors is provided in Appendix C.4.

Compared to existing analyses on diffusion models, Theorem 1 extends result in [3] from exponential-decay step-size to general choices of step-size, and recovers their sharp linear dimension dependence in the discretization error, with minimal assumptions on the target distribution. By assuming two consecutive step sizes are of the same order, we perform asymptotic estimation on the accumulated discretization errors and obtain a bound that depends on the step-size. Such kind of result helps to understand the optimal time schedule, as we will discuss soon.

**Discussion on the choices of time schedule.** Under different choices of step-size, the discretization errors in the denoising diffusion model have the same linear dependence on $d$, but different dependence on $\delta$. The linear dimension dependence improves the results in [4], where $\mathcal{O}(d^2)$ discretization error bounds are proved for different choices of step-size. It is also a extension of the result in [3], where $\mathcal{O}(d)$ discretization error is only proved for the exponential-decay step-size. In fact, Theorem 1 implies that the exponential-decay step-size induces an optimal discretization error up to some constant in term of the inverse early-stopping time $\delta^{-1}$. Detailed discussions on different choices of time schedules is provided in Appendix C.5.

### 3.3 Complexity of ZOD-MC

With the convergence result for DDMC in Theorem 1, we introduce the query complexity bound of ZOD-MC. Our analysis assumes a relaxation of the commonly used gradient-Lipschitz condition on the potential. The formal statement is presented in Corollary 3.1, whose proof is provided in Appendix C.6.

**Assumption 3.2.** *There exists a constant $L > 0$ such that for any $x^* \in \arg\min_{y \in \mathbb{R}^d} V(y)$ and $x \in \mathbb{R}^d$, $V$ satisfies $V(x) - V(x^*) \leq \frac{L}{2} \|x - x^*\|^2$.*

**Corollary 3.1.** *Under the assumptions in Theorem 1 and Assumption 3.2, if we set $T = \frac{1}{2} \ln(\frac{d + \mathrm{m}_2^2}{\varepsilon_{KL}})$, $\gamma_k = \kappa \min(1, T - t_k)$, $\delta = \min(\frac{\varepsilon_{W_2}^2}{d}, \frac{\varepsilon_{W_2}}{\mathrm{m}_2})$, $\kappa = \Theta\left(\frac{T + \ln(\delta^{-1})}{N}\right)$, then to obtain an output (with distribution $q_{t_N}$) in ZOD-MC such that $W_2(p, p_\delta) \lesssim \varepsilon_{W_2}$ and $KL(p_\delta, q_{t_N}) \lesssim \varepsilon_{KL}$, the zeroth-order query complexity is of order*

$$\tilde{\mathcal{O}}\big( \max\big(\tfrac{d + \mathrm{m}_2^2}{\varepsilon_{KL}}, \tfrac{d^2}{\varepsilon_{KL}^2}\big) \varepsilon_{KL}^{-\frac{d-2}{2}} (d + \mathrm{m}_2^2)^{\frac{d-2}{2}} L^{\frac{d}{2}} d^{-1} \big) \max_{0 \leq k \leq N-1} \exp\big(L \|x^*\|^2 + \|x_k\|^2\big), \quad (10)$$

*where the $\tilde{\mathcal{O}}$ hides polylog$(\frac{d + \mathrm{m}_2^2}{\varepsilon_{W_2}})$ factors.*

**Remark 4.** *If we assume WLOG that the minimizer of the potential is at the origin, i.e., $x^* = 0$, and further make reasonable assumptions that $\mathrm{m}_2^2$, $L$ and $\{\|x_k\|^2\}$ are all of order $\mathcal{O}(d)$, where $\{x_k\}$ are the iterates in Algorithm 1, then the query complexity of ZOD-MC is of order $\exp\big(\tilde{\mathcal{O}}(d) \log(\varepsilon_{KL}^{-1})\big)$. Even though this complexity bound has an exponential dimension dependence, it only depends polynomially on the inverse accuracy. Since it applies to any target distribution satisfying Assumptions 3.1 and 3.2, this complexity bound suggests that with the same overall complexity, ZOD-MC can generate samples more accurate than other algorithms in Table 5, for a large class of **low-dimensional non-logconcave** target distributions.*

**Comparison to LMC, RDMC and RSDMC.** When no isoperimetric condition is assumed, we compare convergence for ZOD-MC to convergence for LMC, RDMC and RSDMC.

In the absence of the isoperimetric condition, [1] demonstrated that LMC is capable of producing samples that are close to the target in FI assuming the target potential is smooth. However, FI is a weaker divergence than KL divergence/ Wasserstein-2 distance. It has been observed that, in certain instances, the KL divergence/Wasserstein-2 distance may still be significantly different from zero, despite a minimal FI value. This observation implies that the convergence criteria based on FI may not be as stringent as our result which is based on KL divergence/Wasserstein-2 distance. [26] proved that RDMC produces samples that are $\varepsilon$-close to the target in KL divergence with high probability. Assuming the potential is smooth and a tail-growth condition, the first order oracle complexity is shown to be of order $\exp(\varepsilon^{-1} \log d)$. [27] introduced RSDMC as an acceleration of RDMC. They were able to show that if the potential is smooth, RSDMC produces a sample that is $\varepsilon$-close to the target in KL divergence with high probability. The first order oracle complexity is shown to be of order $\exp(\log^3(d/\varepsilon))$. Compared to RDMC and RSDMC, our result on ZOD-MC doesn't require the potential to be smooth as our Assumption 3.2 is only a growth condition of the potential. This indicates that our convergence result applies to targets with non-smooth, or even discontinuous potentials. Our result in Corollary 3.1 shows the zeroth-order oracle complexity for ZOD-MC is of order $\exp(d \log(\varepsilon^{-1}))$, which achieves a better $\varepsilon$-dependence compared to RDMC and RSDMC, at the price of a worse dimension dependence. This suggests that, for any low-dimensional target, ZOD-MC produces a more accurate sample than RDMC/RSDMC when the overall oracle complexity are the same. Last, zeroth-order queries cost less computationally than first-order queries in practice, which also makes ZOD-MC a more suitable sampling algorithm when the gradients of the potential are hard to compute.

## 4 Experiments

We will demonstrate ZOD-MC on three examples, namely Gaussian mixtures, Gaussian mixtures plus discontinuities, and Müller-Brown which is a highly-nonlinear, nonconvex test problem popular in computational chemistry and material sciences. Multiple Gaussian mixtures will be considered, for showcasing the robustness of our method under worsening isoperimetric properties. The baselines we consider include RDMC [26], RSDMC [27], SLIPS [20], the proximal sampler [37], annealed

importance sample [42], sequential Monte Carlo [15], a parallel tempering approach with MALA proposals [30] and naive unadjusted Langevin Monte Carlo. All the experiments are conducted using a NVIDIA GeForce RTX 4070 Laptop GPU with 8GB of VRAM and Pytorch.

## 4.1 Results for Gaussian Mixtures

**Matched Oracle Complexity.** We modify a 2D Gaussian mixture example frequently considered in the literature to make it more challenging, by making its modes unbalanced with non-isotropic variances, resulting in a highly asymmetrical, multi-modal problem. We include the full details of the parameters in Appendix D. We fix the same oracle complexity (total number of $0^{th}$ and $1^{st}$ order $V$ queries) for different methods, and show the generated samples in Figure 2. Note matching oracle complexity puts our method at a disadvantage, since other techniques require querying the gradient, which results in more function evaluations. Despite this, we see in Figure 1a that our method achieves both the lowest MMD and $W_2$ using the least number of oracle complexity.

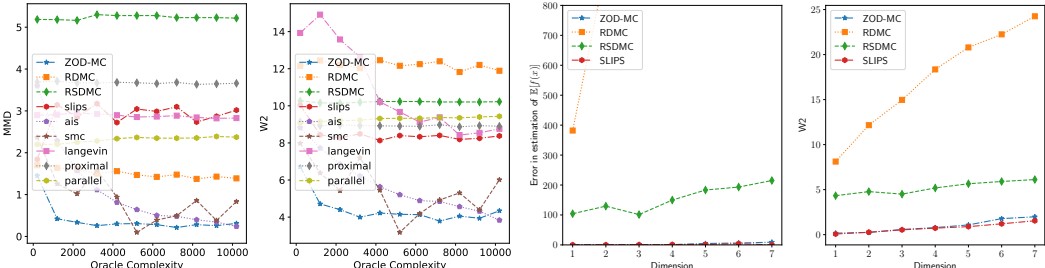

(a) *Sampling accuracy against oracle complexity.* For any fixed oracle complexity, ZOD-MC has the least error both in MMD and $W_2$. Note diffusion based methods do not have an initialization. Thus curves don't start at the same $y$-value.

(b) *Sampling accuracy against dimension* We demonstrate that other diffusion based methods scale poorly with dimension. On the left we plot the error when evaluating statistics of the generated samples and on the right we analyze the $W_2$ metric.

Figure 1: Accuracies of different methods for sampling Gaussian Mixture

**Robustness Against Mode Separation.** Now let's further separate the modes in the mixture to investigate the robustness of our method to increasing nonconvexity/metastability. More precisely, we scale the means of each mode by a constant factor to have a mode located at $(0, R)$; doing so increases the barriers between the modes and exponentially worsens the isoperimetric properties of the target distribution [49]. Figure 4a shows our method is the most insensitive to mode separation. Being the only one that can successfully sample from all modes, as observed in Figure 3, ZOD-MC suffers less from metastability. Note there is still some dependence on mode separation due to the $x_k$ dependence in the complexity bound in Corollary 3.1.

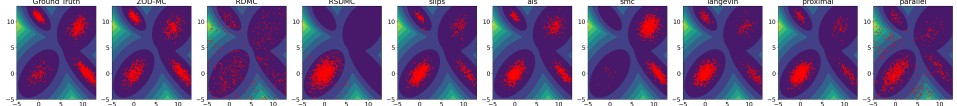

Figure 2: *Sampling from asymmetric, unbalanced Gaussian Mixture.* All diffusion-based methods (ZOD-MC, RDMC, RSDMC) use 2200 oracles per score evaluation. Langevin and the proximal sampler are set to use the same total amount of oracles as diffusion based methods. While other methods suffer from metastability, ZOD-MC correctly samples all modes.

**Dimension Dependence Against Other Diffusion Based Methods.** One drawback of our method, is its bad dimension dependence when compared to diffusion based methods. For instance, RDMC and RSDMC have a dependence of $\exp(\mathcal{O}(\log(d))\tilde{\mathcal{O}}(\varepsilon^{-1}))$ and $\exp(\mathcal{O}(\log^3(d\varepsilon^{-1})))$ respectively, in comparison to our $\exp(\tilde{\mathcal{O}}(d)\mathcal{O}((\log(\varepsilon^{-1}))))$. Despite this theoretical disadvantage, we find empirically that these methods don't scale well with dimension either. To demonstrate this we sample 5 points on the positive quadrant and use them as means for a GMM. We then evaluate statistics on the generated samples and $W_2$ as a function of dimension. We observe in Figure 1b that under a fixed number of function evaluations our method results in the lowest $W_2$. More details are in Appendix D.

**Discontinuous Potentials.** The use of zeroth-order queries allows ZOD-MC to solve problems that would be completely infeasible to first order methods. To demonstrate this, we modify the

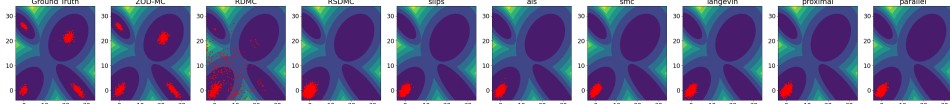

Figure 3: *Gaussian Mixture with further separated modes ($R = 26$)*. ZOD-MC can overcome strengthened metastability and sample from every mode, while other methods are stuck at the mode at the origin, where every method is initialized.

potential in Figure 2. We consider $V(x) + U(x)$ where $U$ is a discontinuous function given by $U(x) = 8\lfloor \|x\| \rfloor \mathbb{1}_{\{5 < \|x\| < 11\}}$ This creates an annulus of much lower probability and a strong potential barrier. In the original problem, the mode centered at the origin was chosen to have the smallest weight (0.1), but adding this discontinuity significantly changes the problem. As observed in Figure 5, our method is still able to correctly sample from the target distribution, while other methods not only continue to suffer from metastability but also fail to see the discontinuities. We quantitatively evaluate the sampling accuracy by using rejection sampling (slow but unbiased) to obtain ground truth samples, and then compute MMD and $W_2$. See Appendix D.2 for details.

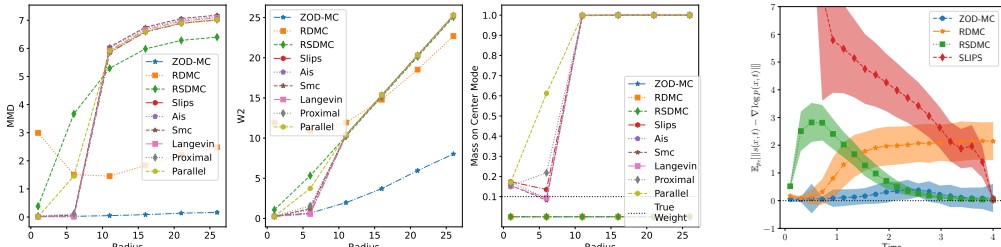

(a) *Sampling accuracy against how separated modes are.* ZOD-MC is the least sensitive to mode separation. The oracle complexity is fixed, independent of how modes are separated.

(b) *Average Score Error as a function of time*. Shaded is the standard deviation of the errors.

Figure 4: *Accuracies of generated samples against dimension and Score Error*. On the right, the result for SLIPS is not directly comparable as it has a different forward process.

**Score Approximation of Diffusion Based Methods.** One explanation of our method's great success in comparison with RDMC and RSDMC is the ability to approximate the score correctly. We select an unbalanced assymetrical 5d GMM and evaluate the average $L^2$ score error between methods. On Figure 4b we show that the best approximations of the score are found by using ZODMC as an estimator as opposed to other methods. Even as $t$ increases and the approximation gets harder we are able to retain accuracy and therefore generate high quality samples.

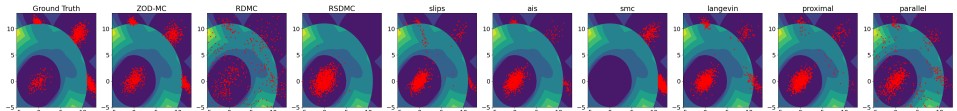

Figure 5: *Generated samples for discontinuous Gaussian Mixture*. Our method can recover the target distribution even under the presence of discontinuities. The same oracle complexity is again used in each method, 3200 per score evaluation in diffusion-based approaches.

## 4.2 Results of Müller Brown Potential

The Müller Brown potential is a toy model for molecular dynamics. Its highly nonlinear potential has 3 modes despite of being the sum of 4 exponentials. The original version has 2 of its modes corresponding to negligible probabilities when compared to the 3rd, which is not good to visualization and comparison across different methods. Thus we consider a balanced version [35] and further translate and dilate $x$ and $y$ so that one of the modes is centered near the origin. The details of the potential can be found in Appendix D.5. Our method is the only one that can correctly sample from all 3 modes as observed in Figure 6 (note they are leveled).

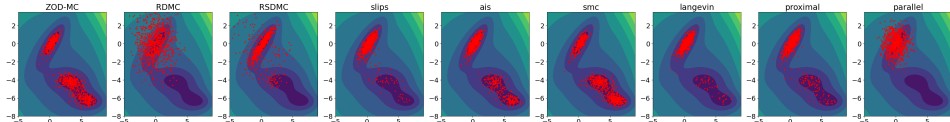

Figure 6: *Generated samples for the Müller Brown potential.* We overlay the generated samples on top of the level curves of $V(x)$. All methods use 1100 oracles.

## Acknowledgments and Disclosure of Funding

The authors are grateful for the partially support by NSF DMS-1847802, Cullen-Peck Scholarship, and GT-Emory Humanity.AI Award. We thank the anonymous reviewers for their helpful comments.

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

## A  Related Works on Zeroth-Order Sampling

The zeroth-order sampling algorithms have been widely studied in the past decades. There is a class of zeroth-order sampling algorithms, including the Ensemble Kalman Inversion [28] and the Ensemble Kalman Sampler [19], that are based on moving a set of easy-to-sample particle according to certain dynamics. However, these methods require (noisy) observations from the target distribution rather than queries of the potential function. Within the zeroth-order sampling algorithms using queries of the potential function, one type of methods make use of the zeroth-order queries to approximate the gradient and apply it to some first-order sampling algorithm [12, 47, 23]. Since it is based on the first-order methods, the analysis of this type of algorithms assumes the target distribution satisfies certain isoperimetric property in general. The other type of methods utilize the zeroth-order queries directly without relating to the gradient. Such methods include the Rejection sampling algorithm, the Metropolized Random Walk (MRW) [41, 46], Ball Walk [39, 17], Hit-and-Run algorithm [2, 40]. The rejection sampling algorithm requires to finding an envelope function which is easy to sample from. This could be difficult. MRW requires sufficient smooth and light tail of the target distribution to mix fast. Ball walk and Hit-and-Run algorithms assume the target distribution is compactly supported. In this paper, we develop a zeroth-order sampling algorithm based on the reverse OU process. Our algorithm does not suffer from the difficulties in the rejection sampling and MRW, and our analysis does not assume isoperimetric property and compact support of the target distribution.

[25] also studies the complexity of a zeroth-order sampling algorithm that combines an approximation technique and rejection sampling. For target distributions that are $m$-differentiable with compact support, [25][Theorem 12] implies a complexity of order $\Omega_d(\varepsilon^{-d/m})$ to reach an $\varepsilon$-accuracy in KL-divergence, where the dimension dependence is implicit. Compare to our result in Corollary 3.1, both complexities are polynomial in $\varepsilon$ and exponential in $d$. Our complexity is smaller for less smooth targets ($m < 2$) while their complexity is smaller for smoother targets ($m > 2$). However, result in [25] only applies to smooth targets ($m > 0$) with compact supports, while our Corollary 3.1 applies to more general target distributions which can be with full support with even discontinuous potentials.

## B  More Details on ZOD-MC

In this section, we provide more details on how rejection sampling (Algorithm 3) in ZOD-MC implements the oracle in DDMC, i.e., generating Monte Carlo samples required in Algorithm 2. ***Construction of an envelope.*** If we have $V^*$ as a minimum value of $V$, then by noting that:

$$-V(z) - \tfrac{1}{2}\frac{\|z - e^t x\|^2}{e^{2t}-1} \leq -V^* - \tfrac{1}{2}\frac{\|z - e^t x\|^2}{e^{2t}-1}, \quad \forall\, z \in \mathbb{R}^d.$$

We are able to construct an envelope for rejection sampling. In particular we propose a samples $z$ from $\mathcal{N}(\cdot\, ; e^t x, e^{2t} - 1)$ and accept proposal $z$ with probability $\exp(-V(z) + V^*)$.

***Sampling from the target distribution.*** Algorithm 3, implements the oracle in Algorithm 2 with $\delta(t) = 0$. When $n(t)$ increases, Algorithm 2 outputs unbiased Monte Carlo score estimators with smaller variance, hence closer to the true score. We will quantify the convergence of DDMC next and consequently demonstrate that ZOD-MC can sample general non-logconcave distributions.

## C  Proofs

### C.1  Properties of the OU-Process

In this section, we introduce and prove some useful properties of the OU-process. Throughout this section, we denote $\{X_t\}_{t\geq 0}$ as the solution of (1) with $p_t := \mathrm{Law}(X_t)$. For any $s, t > 0$, $p_{t|s}$ denotes the conditional probability measure of $X_t$ given the value of $X_s$.

**Proposition C.1.** *(Decay along the OU-Process) Let $\{X_t\}_{t\geq 0}$ be the solution of (1) with $p_t := \mathrm{Law}(X_t)$. Assume that the initial distribution $p$ satisfies Assumption 3.1. Then we have*

$$W_2(p_t, p)^2 \leq (1 - e^{-t})^2 \mathrm{m}_2^2 + (1 - e^{-2t})d, \tag{11}$$

$$and \quad KL(p_t | \gamma^d) \leq \frac{1}{2}\frac{e^{-4t}}{1 - e^{-2t}}d + \frac{1}{2}e^{-2t}\mathrm{m}_2^2. \tag{12}$$

*Proof of Proposition C.1.* The proof for (11) is based on the fact that the solution to (1) can be represented by

$$X_t = e^{-t}X_0 + \sqrt{1 - e^{-2t}}Z, \quad \forall\, t \geq 0. \tag{13}$$

We have

$$W_2(p_t, p)^2 \leq \mathbb{E}\big[\,\|X_t - X_0\|^2\,\big] \leq \mathbb{E}_{(X_0, Z) \sim p \otimes \gamma^d}\big[\|(e^{-t} - 1)X_0 + \sqrt{1 - e^{-2t}}Z\|^2\big]$$
$$= (1 - e^{-t})^2 \mathrm{m}_2^2 + (1 - e^{-2t})d.$$

Next, to prove (12), we have

$$\mathrm{KL}(p_t | \gamma^d) = \mathrm{KL}(\int p_{t|0}(\cdot|y)p(\mathrm{d}y) | \gamma^d(\cdot)) \leq \int \mathrm{KL}(p_{t|0}(\cdot|y)|\gamma^d)p(\mathrm{d}y),$$

where the inequality follows from the convexity of KL divergence. According to (13), $p_{t|0}(\cdot|y)$ is a Gaussian measure with mean $e^{-t}y$ and covariance matrix $(1 - e^{-2t})I_d$. According to [43], we have

$$\mathrm{KL}(p_{t|0}(\cdot|y)|\gamma^d) = \mathrm{KL}(\mathcal{N}(e^{-t}y, (1 - e^{-2t)I_d}) | \mathcal{N}(0, I_d))$$
$$= \frac{1}{2}\big( -d\ln(1 - e^{-2t}) - e^{-2t}d + e^{-2t}\|y\|^2 \big).$$

As a result, we get

$$\mathrm{KL}(p_t | \gamma^d) \leq -\frac{d}{2}\ln(1 - e^{-2t}) - \frac{d}{2}e^{-2t} + \frac{1}{2}e^{-2t}\int \|y\|^2 p(\mathrm{d}y) \leq \frac{e^{-4t}}{2(1 - e^{-2t})}d + \frac{1}{2}e^{-2t}\mathrm{m}_2^2,$$

where the last inequality follows from the fact that $\ln(1 + x) \leq x$ for all $x > 0$. $\qquad\square$

**Proposition C.2.** *(Stochastic Dynamics along the OU-Process) Let $\{X_t\}_{t \geq 0}$ be the solution of (1). Define $m_t(X_t) := \mathbb{E}_{X_0 \sim p_{0|t}(\cdot|X_t)}[X_0]$ and $\Sigma_t(X_t) := \mathrm{Cov}_{X_0 \sim p_{0|t}(\cdot|X_t)}(X_0) = \mathbb{E}_{X_0 \sim p_{0|t}(\cdot|X_t)}[(X_0 - m_t(X_t))^{\otimes 2}]$. Then we have for all $t \geq 0$,*

$$\frac{\mathrm{d}}{\mathrm{d}t}\mathbb{E}\big[\Sigma_t(X_t)\big] = \frac{2e^{-2t}}{(1 - e^{-2t})^2}\mathbb{E}\big[\Sigma_t(X_t)^2\big].$$

The above proposition is known in stochastic localization literature [7] and diffusion model literature [3]. We present its proof for the sake of completeness.

*Proof of Proposition C.2.* For any $T > 0$, from (13), we have the conditional distribution

$$p_{0|t}(\mathrm{d}x|X_t) \propto \exp\big( -\frac{1}{2}\frac{\|X_t - e^{-t}x\|^2}{1 - e^{-2t}} \big)p(\mathrm{d}x),$$

where $\{X_t\}_{0 \leq t \leq T}$ is the solution of (1). Noticing that the solution of (2), $\{\bar{X}_t\}_{0 \leq t \leq T}$ is the reverse process of $\{\bar{X}_t\}_{0 \leq t \leq T}$ and it satisfies $\bar{X}_t = X_{T-t}$ in distribution for all $t \in [0, T]$. Therefore, it suffices to study

$$q_{0|t}(\mathrm{d}x|\bar{X}_t) := Z^{-1}\exp\big( -\frac{1}{2}\frac{\|\bar{X}_t - e^{-(T-t)}x\|^2}{1 - e^{-2(T-t)}} \big)p(\mathrm{d}x)$$
$$= Z_t^{-1}\exp\big( -\frac{1}{2}\frac{e^{-2(T-t)}}{1 - e^{-2(T-t)}}\|x\|^2 + \frac{e^{-(T-t)}}{1 - e^{-2(T-t)}}\langle x, \bar{X}_t\rangle \big)p(\mathrm{d}x)$$
$$:= Z_t^{-1}\exp(h_t(x))p(\mathrm{d}x), \tag{14}$$

where the normalization constant $Z_t = \int_{\mathbb{R}^d}\exp(h_t(x))p(\mathrm{d}x)$. We have $q_{0|t}(\mathrm{d}x|\bar{X}_t) = p_{0|T-t}(\mathrm{d}x|X_{T-t})$ in distribution for all $t \in [0, T]$ and

$$\bar{m}_t(\bar{X}_t) := \mathbb{E}_{X_0 \sim q_{0|t}(\cdot|\bar{X}_t)}[X_0] = \mathbb{E}_{X_0 \sim p_{0|T-t}(\cdot|X_{T-t})}[X_0] = m_{T-t}(X_{T-t}), \tag{15}$$
$$\bar{\Sigma}_t(\bar{X}_t) := \mathrm{Cov}_{X_0 \sim q_{0|t}(\cdot|\bar{X}_t)}(X_0) = \mathrm{Cov}_{X_0 \sim p_{0|T-t}(\cdot|X_{T-t})}(X_0) = \Sigma_{T-t}(X_{T-t}). \tag{16}$$

where the above two identities hold in distribution. For simplicity, we denote $\sigma_t = \sqrt{1 - e^{-2t}}$. Then $h_t(x) = -\frac{1}{2}(\sigma_{T-t}^{-2} - 1)\|x\|^2 + \sigma_{T-t}^{-1}\sqrt{\sigma_{T-t}^{-2} - 1}\langle x, \bar{X}_t\rangle$ is a stochastic process linearly depending on

$\{\bar{X}_t\}_{t\geq 0}$. The conditional measure $\{q_{0|t}(x|\bar{X}_t)\}_{t\geq 0}$ is a measure-valued stochastic process, whose dynamics can be studied by applying Itô's formula. First we have

$$\mathrm{d}h_t(x) = \sigma_{T-t}^{-3}\dot{\sigma}_{T-t}\,\|x\|^2\,\mathrm{d}t + (\sigma_{T-t}^{-2}-1)^{-\frac{1}{2}}\big(-2\sigma_{T-t}^{-4}\dot{\sigma}_{T-t} + \sigma_{T-t}^{-2}\dot{\sigma}_{T-t}\big)\langle x,\bar{X}_t\rangle\mathrm{d}t$$
$$+ \sigma_{T-t}^{-1}(\sigma_{T-t}^{-2}-1)^{\frac{1}{2}}\langle x,\mathrm{d}\bar{X}_t\rangle, \tag{17}$$

$$\mathrm{d}[h(x),h(x)]_t = \sigma_{T-t}^{-2}(\sigma_{T-t}^{-2}-1)\,\|x\|^2\,\mathrm{d}[\bar{X},\bar{X}]_t, \tag{18}$$

Since $\{\bar{X}_t\}_{0\leq t\leq T}$ solves (2), according to Lemma 1, it satisfies that

$$\mathrm{d}\bar{X}_t = \big(\bar{X}_t + 2\mathbb{E}_{x\sim p_{0|T-t}(\cdot|\bar{X}_t)}\big[\frac{e^{-(T-t)}x - \bar{X}_t}{1 - e^{-2(T-t)}}\big]\big)\mathrm{d}t + \sqrt{2}\mathrm{d}\bar{B}_t$$
$$= \big(-\sigma_{T-t}^{-2}(2 - \sigma_{T-t}^{-2})\bar{X}_t + 2(1 - \sigma_{T-t}^2)^{\frac{1}{2}}\bar{m}_t(\bar{X}_t)\big)\mathrm{d}t + \sqrt{2}\mathrm{d}\bar{B}_t. \tag{19}$$

Based on (17), (18) and (19), we have

$$\mathrm{d}Z_t = \int_{\mathbb{R}^d}\exp\big(h_t(x)\big)\big(\mathrm{d}h_t(x) + \frac{1}{2}\mathrm{d}[h(x),h(x)]_t\big)p(\mathrm{d}x)$$
$$= \big(\sigma_{T-t}^{-3}\dot{\sigma}_{T-t} + \sigma_{T-t}^{-2}(\sigma_{T-t}^{-2}-1)\big)\mathbb{E}_{q_{0|t}(\cdot|\bar{X}_t)}\big[\|x\|^2\big]Z_t\mathrm{d}t$$
$$+ (\sigma_{T-t}^{-2}-1)^{-\frac{1}{2}}\big(-2\sigma_{T-t}^{-4}\dot{\sigma}_{T-t} + \sigma_{T-t}^{-2}\dot{\sigma}_{T-t}\big)\langle\bar{m}_t(\bar{X}_t),\bar{X}_t\rangle Z_t\mathrm{d}t$$
$$+ \sigma_{T-t}^{-1}(\sigma_{T-t}^{-2}-1)^{\frac{1}{2}}\langle\bar{m}_t(\bar{X}_t),\mathrm{d}\bar{X}_t\rangle Z_t,$$

$$\text{and}\quad \mathrm{d}\ln Z_t = Z_t^{-1}\mathrm{d}Z_t - \frac{1}{2}Z_t^{-2}\mathrm{d}[Z,Z]_t$$
$$= \big(\sigma_{T-t}^{-3}\dot{\sigma}_{T-t} + \sigma_{T-t}^{-2}(\sigma_{T-t}^{-2}-1)\big)\mathbb{E}_{q_{0|t}(\cdot|\bar{X}_t)}\big[\|x\|^2\big]\mathrm{d}t$$
$$+ (\sigma_{T-t}^{-2}-1)^{-\frac{1}{2}}\big(-2\sigma_{T-t}^{-4}\dot{\sigma}_{T-t} + \sigma_{T-t}^{-2}\dot{\sigma}_{T-t}\big)\langle\bar{m}_t(\bar{X}_t),\bar{X}_t\rangle\mathrm{d}t$$
$$+ \sigma_{T-t}^{-1}(\sigma_{T-t}^{-2}-1)^{\frac{1}{2}}\langle\bar{m}_t(\bar{X}_t),\mathrm{d}\bar{X}_t\rangle$$
$$+ \sigma_{T-t}^{-2}(\sigma_{T-t}^{-2}-1)\,\|\bar{m}_t(\bar{X}_t)\|^2\,\mathrm{d}t. \tag{20}$$

If we define $R_t(\bar{X}_t) = \frac{q_{0|t}(\mathrm{d}x|\bar{X}_t)}{p(\mathrm{d}x)} = Z_t^{-1}\exp(h_t(x))$, then apply Itô's formula again and we have

$$\mathrm{d}R_t(\bar{X}_t) = \mathrm{d}\exp\big(\ln R_t(\bar{X}_t)\big)$$
$$= R_t(\bar{X}_t)\mathrm{d}\big(\ln R_t(\bar{X}_t)\big) + \frac{1}{2}R_t(\bar{X}_t)\mathrm{d}\big[\ln R_t(\bar{X}_t),\ln R_t(\bar{X}_t)\big]$$
$$= R_t(\bar{X}_t)\mathrm{d}h_t(x) - R_t(\bar{X}_t)\mathrm{d}\ln Z_t + \frac{1}{2}R_t(\bar{X}_t)\mathrm{d}\big[h_t(x) - \ln Z_t, h_t(x) - \ln Z_t\big]. \tag{21}$$

Now combine the results in (17), (18), (20) and (21), we can derive the differential equation of $\bar{m}_t(\bar{X}_t)$:

$$\mathrm{d}\bar{m}_t(\bar{X}_t) = \mathrm{d}\int_{\mathbb{R}^d}x R_t(\bar{X}_t)p(\mathrm{d}x)$$
$$= \int_{\mathbb{R}^d}x\big(\mathrm{d}h_t(x) - \mathrm{d}\ln Z_t + \frac{1}{2}\mathrm{d}\big[h_t(x) - \ln Z_t, h_t(x) - \ln Z_t\big]\big)q_{0|t}(\mathrm{d}x|\bar{X}_t)$$
$$= \sigma_{T-t}^{-3}\dot{\sigma}_{T-t}\mathbb{E}_{q_{0|t}(\cdot|\bar{X}_t)}\big[\|x\|^2 x\big]\mathrm{d}t$$
$$+ (\sigma_{T-t}^{-2}-1)^{-\frac{1}{2}}\big(-2\sigma_{T-t}^{-4}\dot{\sigma}_{T-t} + \sigma_{T-t}^{-2}\dot{\sigma}_{T-t}\big)\mathbb{E}_{q_{0|t}(\cdot|\bar{X}_t)}\big[x^{\otimes 2}\big]\bar{X}_t\mathrm{d}t$$
$$+ \sigma_{T-t}^{-1}(\sigma_{T-t}^{-2}-1)^{\frac{1}{2}}\mathbb{E}_{q_{0|t}(\cdot|\bar{X}_t)}\big[x^{\otimes 2}\mathrm{d}\bar{X}_t\big]$$
$$- \sigma_{T-t}^{-3}\dot{\sigma}_{T-t}\mathbb{E}_{q_{0|t}(\cdot|\bar{X}_t)}\big[\|x\|^2\big]\bar{m}_t(\bar{X}_t)\mathrm{d}t$$
$$- (\sigma_{T-t}^{-2}-1)^{-\frac{1}{2}}\big(-2\sigma_{T-t}^{-4}\dot{\sigma}_{T-t} + \sigma_{T-t}^{-2}\dot{\sigma}_{T-t}\big)\bar{m}_t(\bar{X}_t)^{\otimes 2}\bar{X}_t\mathrm{d}t$$
$$- \sigma_{T-t}^{-1}(\sigma_{T-t}^{-2}-1)^{\frac{1}{2}}\bar{m}_t(\bar{X}_t)^{\otimes 2}\mathrm{d}\bar{X}_t$$
$$- \sigma_{T-t}^{-2}(\sigma_{T-t}^{-2}-1)\,\|\bar{m}_t(\bar{X}_t)\|^2\,\bar{m}_t(\bar{X}_t)\mathrm{d}t$$
$$+ \sigma_{T-t}^{-2}(\sigma_{T-t}^{-2}-1)\mathbb{E}_{q_{0|t}(\cdot|\bar{X}_t)}\big[\|x - \bar{m}_t(\bar{X}_t)\|^2 x\big]\mathrm{d}t. \tag{22}$$

Utilize (19) and the definition of $\sigma_t$, all terms with factor $\mathrm{d}t$ in the above equation cancel and (22) can be simplified as

$$
\begin{aligned}
\mathrm{d}\bar{m}_t(\bar{X}_t) &= \frac{\sqrt{2}e^{-(T-t)}}{1 - e^{-2(T-t)}} \mathbb{E}_{q_{0|t}(\cdot|\bar{X}_t)}\big[x \otimes (x - \bar{m}_t(\bar{X}_t))\mathrm{d}B_t\big] \\
&= \frac{\sqrt{2}e^{-(T-t)}}{1 - e^{-2(T-t)}} \bar{\Sigma}_t(\bar{X}_t)\mathrm{d}\bar{B}_t.
\end{aligned}
\tag{23}
$$

Last, we derive the differential equation that $\mathbb{E}_{X_t \sim p_t}\big[\Sigma_t(X_t)\big]$ satisfies. Let $f(t) := \mathbb{E}_{X_t \sim p_t}\big[\Sigma_t(X_t)\big]$ and $g(t) := \mathbb{E}_{X_t \sim p_t}\big[\Sigma_t(X_t)^2\big]$ be two deterministic functions on $[0, T]$. According to (16) and (23), we have

$$
\begin{aligned}
\frac{\mathrm{d}}{\mathrm{d}t} f(T - t) &= \frac{\mathrm{d}}{\mathrm{d}t} \mathbb{E}_{X_{T-t} \sim p_{T-t}} \big[\Sigma_{T-t}(X_{T-t})\big] \\
&= \frac{\mathrm{d}}{\mathrm{d}t} \mathbb{E}_{\bar{X}_t \sim p_{T-t}} \big[\bar{\Sigma}_t(\bar{X}_t)\big] \\
&= \frac{\mathrm{d}}{\mathrm{d}t} \mathbb{E}_{\bar{X}_t \sim p_{T-t}} \big[\mathbb{E}_{q_{0|t}(\cdot|\bar{X}_t)}[x^{\otimes 2}] - \bar{m}_t(\bar{X}_t)^{\otimes 2}\big] \\
&= \frac{\mathrm{d}}{\mathrm{d}t} \mathbb{E}_{x \sim p}[x^{\otimes 2}] - \frac{\mathrm{d}}{\mathrm{d}t} \mathbb{E}_{X_t \sim p_t}\big[\bar{m}_t(\bar{X}_t)^{\otimes 2}\big] \\
&= -\frac{2e^{-2(T-t)}}{(1 - e^{-2(T-t)})^2} \mathbb{E}_{\bar{X}_t \sim p_{T-t}} \big[\bar{\Sigma}_t(\bar{X}_t)^2\big] \\
&= -\frac{2e^{-2(T-t)}}{(1 - e^{-2(T-t)})^2} g(T - t).
\end{aligned}
$$

where the last inequality follows from the Itô isometry. Proposition C.2 is then proved by reverse the time in $f$ and $g$. $\qquad\square$

## C.2  Proofs of Section 3.1

*Proof of Lemma 1.* Based on (13), we have $p_t = (e^{-t})_{\#p} * (\sqrt{1 - e^{-2t}})_{\#\gamma^d}$ where $(e^{-t})_{\#p}$ is the pushforward measure of $p$ via map $x \in \mathbb{R}^d \mapsto e^{-t}x$ and $(\sqrt{1 - e^{-2t}})_{\#\gamma^d}$ is the pushforward measure of $\gamma^d$ via map $x \in \mathbb{R}^d \mapsto \sqrt{1 - e^{-2t}}x \in \mathbb{R}^d$. The pushforward measures $(e^{-t})_{\#p}$ and $(\sqrt{1 - e^{-2t}})_{\#\gamma^d}$ can be written as

$$
(e^{-t})_{\#p}(\mathrm{d}x) = e^{td}p(e^t\mathrm{d}x) \quad \text{and}
$$

$$
(\sqrt{1 - e^{-2t}})_{\#\gamma^d}(\mathrm{d}x) = \big(2\pi(1 - e^{-2t})\big)^{-\frac{d}{2}} \exp\big(-\frac{\|x\|^2}{2(1 - e^{-2t})}\big)\mathrm{d}x,
$$

respectively. Therefore the score function $\nabla \ln p_t(x)$ can be written as

$$
\begin{aligned}
\nabla \ln p_t(x) &= p_t(x)^{-1} e^{td} \big(2\pi(1 - e^{-2t})\big)^{-\frac{d}{2}} \nabla_x \int \exp\big(-\frac{\|x - z\|^2}{2(1 - e^{-2t})}\big) p(e^t\mathrm{d}z) \\
&= p_t(x)^{-1} \big(2\pi(1 - e^{-2t})\big)^{-\frac{d}{2}} \nabla_x \int \exp\big(-\frac{\|x - e^{-t}z\|^2}{2(1 - e^{-2t})}\big) p(\mathrm{d}z) \\
&= \int \frac{x - e^{-t}z}{1 - e^{-2t}} \frac{p_{t|0}(x|z)p(\mathrm{d}z)}{p_t(x)} \\
&= \int \frac{x - e^{-t}z}{1 - e^{-2t}} p_{0|t}(\mathrm{d}z|x),
\end{aligned}
$$

where the last step follows from the Bayesian rule and

$$
p_{0|t}(\cdot|x) \propto p_{t|0}(x|\cdot)p(\cdot) \propto \exp\big(-V(\cdot) - \frac{1}{2}\frac{\|x - e^{-t}\cdot\|^2}{1 - e^{-2t}}\big).
$$

$\qquad\square$

## C.3 Proofs of Section 3.2

*Proof of Proposition 3.1.* With the score estimator given in Algorithm 2, we have

$$\mathbb{E}\big[\,\|\nabla\ln p_t(X_t)-s(t,X_t)\|^2\,\big]$$

$$=\mathbb{E}_{x\sim p_t}\Big[\|\nabla\ln p_t(x)-\frac{1}{n(t)}\sum_{i=1}^{n(t)}\frac{e^{-t}z_{t,i}-x}{1-e^{-2t}}\|^2\Big]$$

$$=\mathbb{E}_{x\sim p_t}\Big[\|\nabla\ln p_t(x)-\frac{1}{n(t)}\sum_{i=1}^{n(t)}\frac{e^{-t}x_{t,i}-x}{1-e^{-2t}}+\frac{1}{n(t)}\sum_{i=1}^{n(t)}\frac{x_{t,i}-z_{t,i}}{1-e^{-2t}}\|^2\Big],$$

where $\{x_{t,i}\}_{i=1}^{n(t)}$ is a sequence of i.i.d. samples following $p_{0|t}(\cdot|x)$ that are chosen such that $\mathbb{E}[\|x_{t,i}-z_{t,i}\|^2\,|x]=W_2(p_{0|t}(\cdot|x),\mathrm{Law}(z_{t,i}))$ for all $t>0$ and $i=1,2,\cdots,n(t)$. Based on Lemma 1, $\{\frac{e^{-t}x_{t,i}-x}{1-e^{-2t}}\}_{i=1}^{n(t)}$ is a sequence of unbiased i.i.d. Monte Carlo estimator of $\nabla\ln p_t(x)$ for all $t>0$ and $x\in\mathbb{R}^d$. Therefore, we get

$$\mathbb{E}\big[\,\|\nabla\ln p_t(X_t)-s(t,X_t)\|^2\,\big]$$

$$=\mathbb{E}_{x\sim p_t}\Big[\|\nabla\ln p_t(x)-\frac{1}{n(t)}\sum_{i=1}^{n(t)}\frac{e^{-t}x_{t,i}-x}{1-e^{-2t}}+\|^2\Big]+\mathbb{E}\Big[\|\frac{1}{n(t)}\sum_{i=1}^{n(t)}\frac{x_{t,i}-z_{t,i}}{1-e^{-2t}}\|^2\Big]$$

$$=\underbrace{\frac{1}{n(t)^2}\sum_{i=1}^{n(t)}\mathbb{E}_{x\sim p_t}\Big[\|\frac{e^{-t}x_{t,i}-x}{1-e^{-2t}}-\mathbb{E}_{x_{t,i}\sim p_{0|t}(\cdot|x)}[\frac{e^{-t}x_{t,i}-x}{1-e^{-2t}}]\|^2\Big]}_{N_1}$$

$$+\underbrace{\frac{e^{-2t}}{(1-e^{-2t})^2}\frac{1}{n(t)^2}\sum_{i,j=1}^{n(t)}\mathbb{E}\big[\langle x_{t,i}-z_{t,i},x_{t,j}-z_{t,j}\rangle\big]}_{N_2}.$$

The first term in the above equation, $N_1$, is related to the covariance of $p_{0|t}(\cdot|X_t)$, which is studied in Proposition C.2. We have

$$N_1=\frac{e^{-2t}}{(1-e^{-2t})^2}\frac{1}{n(t)^2}\sum_{i=1}^{n(t)}\mathbb{E}_{x\sim p_t}\big[\mathrm{trace}\big(\mathrm{Cov}_{x_{t,i}\sim p_{0|t}(\cdot|x)}(x_{t,i})\big)\big]$$

$$=\frac{e^{-2t}}{(1-e^{-2t})^2}\frac{1}{n(t)}\mathbb{E}_{x\sim p_t}\big[\mathrm{trace}\big(\Sigma_t(x)\big)\big]$$

$$\leq\frac{e^{-2t}}{(1-e^{-2t})^2}\frac{1}{n(t)}\mathbb{E}_{x\sim\gamma^d}\big[\mathrm{trace}\big(\Sigma_\infty(x)\big)\big]$$

$$=\frac{e^{-2t}}{(1-e^{-2t})^2}\frac{1}{n(t)}\mathrm{Cov}_p(x),$$

where the inequality follows from Proposition C.2 indicating that $t\mapsto\mathbb{E}_{x\sim p_t}\big[\mathrm{trace}\big(\Sigma_t(x)\big)\big]$ is a increasing function.

The second term $N_2$ characterize the bias from the Monte Carlo samples and the bias can be measured by the Wasserstein-2 distance:

$$N_2\leq\frac{e^{-2t}}{(1-e^{-2t})^2}\frac{1}{n(t)^2}\sum_{i,j=1}^{n(t)}\mathbb{E}\big[\|x_{t,i}-z_{t,i}\|^2\big]^{\frac{1}{2}}\mathbb{E}\big[\|x_{t,j}-z_{t,j}\|^2\big]^{\frac{1}{2}}$$

$$=\frac{e^{-2t}}{(1-e^{-2t})^2}\frac{1}{n(t)^2}\sum_{i,j=1}^{n(t)}\mathbb{E}_{x\sim p_t}[W_2(\mathrm{Law}(z_{t,i}),p_{0|t}(\cdot|x))]\mathbb{E}_{x\sim p_t}[W_2(\mathrm{Law}(z_{t,j}),p_{0|t}(\cdot|x))]$$

$$\leq\frac{e^{-2t}}{(1-e^{-2t})^2}\delta(t)^2.$$

(4) follows from the estimation on $N_1$ and $N_2$. $\qquad\square$

## C.4 Proof of Theorem 1

In this section, we introduce the proof of our main convergence results, Theorem 1. Recall that in the convergence result in Theorem 1, three types of errors appear in the upper bound: the initialization

error, the discretization error and the score estimation error. Our proof compares the trajectory of $\{\tilde{X}_t\}_{0\leq t\leq T}$ that solves (8) and the trajectory of $\{\bar{X}_t\}_{0\leq t\leq T}$ that solves (2). We denote the path measures of $\{\bar{X}_t\}_{0\leq t\leq T}$ and $\{\tilde{X}_t\}_{0\leq t\leq T}$ by $P^{p_T}$, and $Q^{\gamma^d}$, respectively. Next, we introduce a high level idea on how the three types of errors are handled .

1. **Initialization error:** the initialization error comes from the comparison between $\{\tilde{X}_t\}_{0\leq t\leq T}$ and $\{\tilde{X}_t^{p_T}\}_{0\leq t\leq T}$. To characterize this error, we introduce the intermediate process $\{\tilde{X}_t^{p_T}\}_{0\leq t\leq T}$

$$\mathrm{d}\tilde{X}_t^{p_T} = (\tilde{X}_t^{p_T} + 2s(T - t_k, \tilde{X}_{t_k}^{p_T}))\mathrm{d}t + \sqrt{2}\mathrm{d}\tilde{B}_t, \quad \tilde{X}_0 \sim p_T, \ t \in [t_k, t_{k+1}), \quad (24)$$

in (24) and denote the path measure of $\{\tilde{X}_t^{p_T}\}_{0\leq t\leq T}$ by $Q^{p_T}$. Both processes are driven by the estimated scores and only the initial conditions are different. We factor out the initialization error from $\mathrm{KL}(p_\delta|q_{t_N})$ by the following argument:

$$\begin{aligned}
\mathrm{KL}(p_\delta|q_{t_N}) &= \mathrm{KL}(p_\delta|q_{T-\delta}) \\
&\leq \int \ln \frac{\mathrm{d}P^{p^T}}{\mathrm{d}Q^{\gamma^d}}\mathrm{d}P^{p_T} = \int \ln \big(\frac{\mathrm{d}P^{p^T}}{\mathrm{d}Q^{p_T}}\frac{\mathrm{d}Q^{p_T}}{\mathrm{d}Q^{\gamma^d}}\big)\mathrm{d}P^{p_T} \\
&= \mathrm{KL}(P^{p_T}|Q^{p_T}) + \int \ln \frac{\mathrm{d}Q^{p_T}}{\mathrm{d}Q^{\gamma^d}}\mathrm{d}P^{p_T} \\
&= \mathrm{KL}(P^{p_T}|Q^{p_T}) + \mathrm{KL}(p_T|\gamma^d),
\end{aligned}$$

where the inequality follows from the data processing inequality and the last identity follows from the fact that $\frac{\mathrm{d}Q^{p_T}}{\mathrm{d}Q^{\gamma^d}} = \frac{\mathrm{d}Q_0^{p_T}}{\mathrm{d}Q_0^{\gamma^d}} = \frac{\mathrm{d}p_T}{\mathrm{d}\gamma^d}$, which is true because the processes $\{\tilde{X}_t\}_{0\leq t\leq T}$ and $\{\tilde{X}_t^{p_T}\}_{0\leq t\leq T}$ have the same transition kernel function. $\mathrm{KL}(p_T|\gamma^d)$ is the initialization error and it is bounded based on (12) in Proposition C.1.

2. **Discretization error:** the dicretization error arises from the evaluations of the scores at the discrete times. We factor out the discretization error from the $\mathrm{KL}(P^{p_T}|Q^{p_T})$ via the Girsanov's Theorem.

$$\begin{aligned}
\mathrm{KL}(P^{p_T}|Q^{p_T}) &\leq \sum_{k=0}^{N-1} \int_{t_k}^{t_{k+1}} \mathbb{E}_{P^{p_T}}\big[\big\|\nabla \ln p_{T-t}(\bar{X}_t) - s(T - t_k, \bar{X}_{t_k})\big\|^2\big]\mathrm{d}t \\
&\lesssim \underbrace{\sum_{k=0}^{N-1} \int_{t_k}^{t_{k+1}} \mathbb{E}_{P^{p_T}}\big[\big\|\nabla \ln p_{T-t}(\bar{X}_t) - \nabla \ln p_{T-t_k}(\bar{X}_{t_k})\big\|^2\big]\mathrm{d}t}_{\text{discretization error}} \\
&\quad + \underbrace{\sum_{k=0}^{N-1} \gamma_k\mathbb{E}_{P^{p_T}}\big[\big\|\nabla \ln p_{T-t_k}(\bar{X}_{t_k}) - s(T - t_k, \bar{X}_{t_k})\big\|^2\big]}_{\text{score estimation error}}
\end{aligned}$$

We bound the discretization error term in the above equation by checking the dynamical properties of the process $\{\nabla \ln p_t(\bar{X}_t)\}_{0\leq t\leq T}$. Similar approach was used in the analysis of denoising diffusion models, see [3]. For the sake of completeness, we include the proof in Appendix C.7.

3. **Score estimation error:** as discussed in the discretization error, the score estimation error is the accumulation of the $L^2$-error between the true score and score estimator at the time schedules $\{T - t_k\}_{k=0}^{N-1}$. In the analysis of denoising diffusion models, [5, 4, 3], it is usually assumed that such a $L^2$ score error is small. In this paper, we consider to do sampling via the Reverse OU-process and score estimation. One of our main contribution is that we prove the $L^2$ score error can be guaranteed small for the class of Monte Carlo score estimators given in Algorithm 2. The $L^2$ score error upper bound is stated in Proposition 3.1.

*Proof of Theorem 1.* First we can decompose $\mathrm{KL}(p_\delta|q_{t_N})$ into summation of the three types of error.

$$\mathrm{KL}(p_\delta|q_{t_N}) = \mathrm{KL}(p_\delta|q_{T-\delta})$$

$$\leq \int \ln \frac{\mathrm{d}P^{p^T}}{\mathrm{d}Q^{\gamma^d}} \mathrm{d}P^{p_T} = \int \ln \big( \frac{\mathrm{d}P^{p^T}}{\mathrm{d}Q^{p_T}} \frac{\mathrm{d}Q^{p_T}}{\mathrm{d}Q^{\gamma^d}} \big) \mathrm{d}P^{p_T}$$

$$= \mathrm{KL}(P^{p_T}|Q^{p_T}) + \int \ln \frac{\mathrm{d}Q^{p_T}}{\mathrm{d}Q^{\gamma^d}} \mathrm{d}P^{p_T}$$

$$= \mathrm{KL}(P^{p_T}|Q^{p_T}) + \mathrm{KL}(p_T|\gamma^d)$$

$$\leq \mathrm{KL}(p_T|\gamma^d) + \sum_{k=0}^{N-1} \int_{t_k}^{t_{k+1}} \mathbb{E}_{P^{p_T}} \big[ \big\| \nabla \ln p_{T-t}(\bar{X}_t) - s(T-t_k, \bar{X}_{t_k}) \big\|^2 \big] \mathrm{d}t$$

$$\lesssim \underbrace{\mathrm{KL}(p_T|\gamma^d)}_{\text{initialization error}} + \underbrace{\sum_{k=0}^{N-1} \int_{t_k}^{t_{k+1}} \mathbb{E}_{P^{p_T}} \big[ \big\| \nabla \ln p_{T-t}(\bar{X}_t) - \nabla \ln p_{T-t_k}(\bar{X}_{t_k}) \big\|^2 \big] \mathrm{d}t}_{\text{discretization error}}$$

$$+ \underbrace{\sum_{k=0}^{N-1} \gamma_k \mathbb{E}_{P^{p_T}} \big[ \big\| \nabla \ln p_{T-t_k}(\bar{X}_{t_k}) - s(T-t_k, \bar{X}_{t_k}) \big\|^2 \big]}_{\text{score estimation error}},$$

where the first inequality follows from the data processing inequality. The second inequality follows from Girsanov's theorem and [6, Section 3.1]. According to Proposition C.1 and the assumption that $T > 1$, the initialization error satisfies

$$\mathrm{KL}(p_T|\gamma^d) \leq \frac{1}{2} \frac{e^{-4T}}{1 - e^{-2T}} d + \frac{1}{2} e^{-2T} \mathrm{m}_2^2 \lesssim (d + \mathrm{m}_2^2) e^{-2T}. \tag{25}$$

According to Lemma 2, the discretization error satisfies

$$\sum_{k=0}^{N-1} \int_{t_k}^{t_{k+1}} \mathbb{E}_{P^{p_T}} \big[ \big\| \nabla \ln p_{T-t}(\bar{X}_t) - \nabla \ln p_{T-t_k}(\bar{X}_{t_k}) \big\|^2 \big] \mathrm{d}t$$

$$\lesssim d \sum_{k=0}^{N} \frac{\gamma_k^2}{(1 - e^{-2(T-t_k)})^2}$$

$$+ \sum_{k=0}^{N} \frac{e^{-2(T-t_k)} \gamma_k}{(1 - e^{-2(T-t_k)})^2} \Big( \mathbb{E} \big[ \mathrm{trace} \big( \Sigma_{T-t_k}(X_{T-t_k}) \big) \big] - \mathbb{E} \big[ \mathrm{trace} \big( \Sigma_{T-t_{k+1}}(X_{T-t_{k+1}}) \big) \big] \Big). \tag{26}$$

Reordering the summation in the second term and we have

$$\sum_{k=0}^{N} \frac{e^{-2(T-t_k)} \gamma_k}{(1 - e^{-2(T-t_k)})^2} \Big( \mathbb{E} \big[ \mathrm{trace} \big( \Sigma_{T-t_k}(X_{T-t_k}) \big) \big] - \mathbb{E} \big[ \mathrm{trace} \big( \Sigma_{T-t_{k+1}}(X_{T-t_{k+1}}) \big) \big] \Big)$$

$$\leq \sum_{k=1}^{N-1} \Big( \frac{e^{-2(T-t_k)} \gamma_k}{(1 - e^{-2(T-t_k)})^2} - \frac{e^{-2(T-t_{k-1})} \gamma_{k-1}}{(1 - e^{-2(T-t_{k-1})})^2} \Big) \mathbb{E} \big[ \mathrm{trace} \big( \Sigma_{T-t_k}(X_{T-t_k}) \big) \big]$$

$$+ \frac{e^{-2T} \gamma_0}{(1 - e^{-2T})^2} \mathbb{E} \big[ \mathrm{trace} \big( \Sigma_T(X_T) \big) \big]$$

$$\lesssim \sum_{k=1}^{N-1} \frac{\gamma_k \gamma_{k-1}^2}{(1 - e^{-2(T-t_k)})^2 (1 - e^{-2(T-t_{k-1})})^2} \mathbb{E} \big[ \mathrm{trace} \big( \Sigma_{T-t_k}(X_{T-t_k}) \big) \big]$$

$$+ \frac{e^{-2T} \gamma_0}{(1 - e^{-2T})^2} \mathbb{E} \big[ \mathrm{trace} \big( \Sigma_T(X_T) \big) \big]. \tag{27}$$

Recall that $\Sigma_t(X_t) = \mathrm{Cov}(X_0|X_t)$ for all $0 \leq t \leq T$. We have

$$\mathbb{E} \big[ \mathrm{trace} \big( \Sigma_t(X_t) \big) \big] = \mathbb{E} \big[ \mathrm{Cov}(X_0|X_t) \big] \leq \mathbb{E} \big[ \|X_0\|^2 \big] \leq \mathrm{m}_2^2, \tag{28}$$

$$\text{and} \quad \mathbb{E} \big[ \mathrm{trace} \big( \Sigma_t(X_t) \big) \big] = \mathbb{E} \big[ \mathrm{Cov}(X_0|X_t) \big] = \mathbb{E} \big[ \mathrm{Cov}(X_0 - e^t X_t|X_t) \big]$$

$$\leq \mathbb{E} \big[ \mathbb{E} \big[ \|X_0 - e^t X_t\|^2 |X_t \big] \big]$$

$$= (e^{2t} - 1)d. \tag{29}$$

where the last identity follows from (13). (29) and (28) implies that $\mathbb{E}\big[\text{trace}\big(\Sigma_t(X_t)\big)\big] \lesssim (1-e^{-2t})(d+\mathrm{m}_2^2)$ for all $0 \le t \le T$. Therefore, from (26) and (27), the overall discretization error can be bounded as

$$\sum_{k=0}^{N-1}\int_{t_k}^{t_{k+1}} \mathbb{E}_{P^{p_T}}\big[\,\big\|\nabla \ln p_{T-t}(\bar{X}_t) - \nabla \ln p_{T-t_k}(\bar{X}_{t_k})\big\|^2\,\big]\mathrm{d}t$$

$$\lesssim \sum_{k=0}^{N-1}\frac{d\gamma_k^2}{(1-e^{-2(T-t_k)})^2} + \sum_{k=1}^{N-1}\frac{(d+\mathrm{m}_2^2)\gamma_k\gamma_{k-1}^2}{(1-e^{-2(T-t_k)})(1-e^{-2(T-t_{k-1})})^2} + \frac{\mathrm{m}_2^2 e^{-2T}\gamma_0}{(1-e^{-2T})^2}. \qquad (30)$$

Last, according to Proposition 3.1, the score estimation error satisfies

$$\sum_{k=0}^{N-1}\gamma_k\mathbb{E}_{P^{p_T}}\big[\,\big\|\nabla \ln p_{T-t_k}(\bar{X}_{t_k}) - s(T-t_k,\bar{X}_{t_k})\big\|^2\,\big]$$

$$\le \sum_{k=0}^{N-1}\frac{\gamma_k e^{-2(T-t_k)}}{(1-e^{-2(T-t_k)})^2}\bigg(\delta(T-t_k)^2 + \frac{\mathrm{m}_2^2}{n(T-t_k)}\bigg), \qquad (31)$$

and (9) follows from (25), (30) and (31). $\qquad\square$

## C.5   Discussion on the Step-size

In this section, we first state error bounds of DDMC under different choices of step-size. Then we provide the detailed calculations. Last we compare our results in Theorem 1 to existing results on convergence of denoising diffusion models.

In the following discussion, we assume $\delta(T-t_k)^2 \le d\gamma_k e^{2(T-t_k)}$ and $n(T-t_k) \ge \gamma_k^{-1}e^{-2(T-t_k)}$ for all $k = 0, 1, \cdots, N-1$, so that the score estimation error is dominated by the discretization error. For different choices of step-size, we discuss the parameter dependence of the error bound in (9) under the assumptions on $\delta(t)$ and $n(t)$.

1. **constant step-size:** the constant step-size is widely considered in sampling algorithms and denoising diffusion generative models. It requires $\gamma_k = \gamma$ for all $0 \le k \le N-1$. Then

$$\text{KL}(p_\delta|q_{t_N}) \lesssim (d+\mathrm{m}_2^2)e^{-2T} + \frac{(d+\mathrm{m}_2^2)T^2}{N^2}(T+\delta^{-2}) + \frac{dT}{N}(T+\delta^{-1}).$$

2. **linear step-size:** the linear step-size is considered by [6] as an interpretation of the uniform discretization of a diffusion model with non-constant diffusion coefficient [52]. It requires $t_k = T - (\delta + (N-k)\gamma)^2$ with $\gamma = \frac{\sqrt{T}-\delta}{N}$ for all $0 \le k \le N-1$. Then

$$\text{KL}(p_\delta|q_{t_N}) \lesssim (d+\mathrm{m}_2^2)e^{-2T} + \frac{(d+\mathrm{m}_2^2)T}{N^2}(T^2+\delta^{-1}) + \frac{dT^{\frac{1}{2}}}{N}(T^{\frac{3}{2}}+\delta^{-\frac{1}{2}}).$$

3. **exponential-decay step-size:** the exponential-decay step-size is considered to be optimal in SGMs [6, 3]. It requires $\gamma_k = \kappa \min(1, T-t_k)$ for some $\kappa \in (0,1)$. Then

$$\text{KL}(p_\delta|q_{t_N}) \lesssim (d+\mathrm{m}_2^2)e^{-2T} + \frac{(d+\mathrm{m}_2^2)}{N^2}\big(T+\ln(\tfrac{1}{\delta})\big)^3 + \frac{d}{N}\big(T+\ln(\tfrac{1}{\delta})\big)^2.$$

The purple terms are denoting the discretization errors. For all of the above choices of step-size, the error bounds have the same linear dimension dependence and different dependence on the early stopping parameter $\delta$. Next, we provide a detailed calculation of these error bounds and a derivation of optimal $\delta$-dependence.

1. **constant step-size:** when $\gamma_k = \gamma$ for all $k = 0, 1, \cdots, N-1$, we have $T - t_k = \delta + (N-k)\gamma$ and

$$\frac{\gamma_k}{1-e^{-2(T-t_k)}} = \begin{cases} \Theta(\gamma), & \text{if } T-t_k > 1, \\ \Theta\big(\frac{\gamma}{T-t_k}\big), & \text{if } T-t_k < 1. \end{cases}$$

Therefore

$$\sum_{k=1}^{N-1} \frac{(d+\mathrm{m}_2^2)\gamma_k\gamma_{k-1}^2}{(1-e^{-2(T-t_k)})(1-e^{-2(T-t_{k-1})})^2} + \sum_{k=0}^{N-1} \frac{d\gamma_k^2}{(1-e^{-2(T-t_k)})^2}$$

$$= \Theta\left( \sum_{1<T-t_k<T} (d+\mathrm{m}_2^2)\gamma^3 + d\gamma^2 + \sum_{\delta<T-t_k<1} \frac{(d+\mathrm{m}_2^2)\gamma^3}{(T-t_k)(T-t_{k-1}^2)} + \frac{d\gamma^2}{(T-t_k)^2} \right)$$

$$= \Theta\left( \frac{(d+\mathrm{m}_2^2)T^3}{N^2} + \frac{dT^2}{N} + (d+\mathrm{m}_2^2)\gamma^2 \int_\delta^1 t^{-3}\mathrm{d}t + d\gamma \int_\delta^1 t^{-2}\mathrm{d}t \right)$$

$$= \Theta\left( \frac{(d+\mathrm{m}_2^2)T^3}{N^2} + \frac{dT^2}{N} + \frac{(d+\mathrm{m}_2^2)T^2}{N^2\delta^2} + \frac{dT}{N\delta} \right).$$

2. **linear step-size:** when $T - t_k = (\delta + (N-k)\gamma)^2$ with $\gamma = \frac{\sqrt{T}-\delta}{N}$, we have $\gamma_k = (2\delta + (2N - 2k-1)\gamma)\gamma = \Theta(\sqrt{T-t_k}\gamma)$ and

$$\frac{\gamma_k}{1-e^{-2(T-t_k)}} = \begin{cases} \Theta(\gamma\sqrt{T-t_k}), & \text{if } T-t_k > 1, \\ \Theta(\dfrac{\gamma}{\sqrt{T-t_k}}), & \text{if } T-t_k < 1. \end{cases}$$

Therefore

$$\sum_{k=1}^{N-1} \frac{(d+\mathrm{m}_2^2)\gamma_k\gamma_{k-1}^2}{(1-e^{-2(T-t_k)})(1-e^{-2(T-t_{k-1})})^2} + \sum_{k=0}^{N-1} \frac{d\gamma_k^2}{(1-e^{-2(T-t_k)})^2}$$

$$= \Theta\left( \sum_{1<T-t_k<T} (d+\mathrm{m}_2^2)\gamma^3\sqrt{T-t_k}(T-t_{k-1}) + d\gamma^2(T-t_{k-1}) \right)$$

$$+ \Theta\left( \sum_{\delta<T-t_k<1} \frac{(d+\mathrm{m}_2^2)\gamma^3}{\sqrt{T-t_k}(T-t_{k-1})} + \frac{d\gamma^2}{T-t_k} \right)$$

$$= \Theta\left( \gamma^2(d+\mathrm{m}_2^2)\int_1^T t\mathrm{d}t + \gamma d\int_1^T t^{\frac{1}{2}}\mathrm{d}t + \gamma^2(d+\mathrm{m}_2^2)\int_\delta^1 t^{-2}\mathrm{d}t + \gamma d\int_\delta^1 t^{-\frac{3}{2}}\mathrm{d}t \right)$$

$$= \Theta\left( \frac{(d+\mathrm{m}_2^2)T^3}{N^2} + \frac{dT^2}{N} + \frac{(d+\mathrm{m}_2^2)T}{N^2\delta} + \frac{dT^{\frac{1}{2}}}{N\delta^{\frac{1}{2}}} \right).$$

3. **exponential-decay step-size:** when $\gamma_k = \kappa\min(1, T-t_k)$ with $\kappa = \frac{T+\ln(1/\delta)}{N}$, we have

$$\frac{\gamma_k}{1-e^{-2(T-t_k)}} = \begin{cases} \Theta(\kappa), & \text{if } T-t_k > 1, \\ \Theta(\kappa), & \text{if } T-t_k < 1. \end{cases}$$

Therefore

$$\sum_{k=1}^{N-1} \frac{(d+\mathrm{m}_2^2)\gamma_k\gamma_{k-1}^2}{(1-e^{-2(T-t_k)})(1-e^{-2(T-t_{k-1})})^2} + \sum_{k=0}^{N-1} \frac{d\gamma_k^2}{(1-e^{-2(T-t_k)})^2}$$

$$= \Theta\left( \sum_{1<T-t_k<T} (d+\mathrm{m}_2^2)\kappa^3 + d\kappa^2 + \sum_{\delta<T-t_k<1} (d+\mathrm{m}_2^2)\kappa^2\gamma_k(T-t_k)^{-1} + d\kappa\gamma_k(T-t_k)^{-1} \right)$$

$$= \Theta\left( (d+\mathrm{m}_2^2)\kappa^3 N + d\kappa^2 N + \kappa^2(d+\mathrm{m}_2^2)\int_\delta^1 t^{-1}\mathrm{d}t + \kappa d\int_\delta^1 t^{-1}\mathrm{d}t \right)$$

$$= \Theta\left( \frac{(d+\mathrm{m}_2^2)(T+\ln(1/\delta))^3}{N^2} + \frac{d(T+\ln(1/\delta))^2}{N} \right).$$

4. **Optimality of the exponential step-size:** assuming that $\gamma_k = \Theta(\gamma_{k-1})$ for all $k = 0, 1, \cdots, N-1$, the exponential step-size actually provides the optimal order estimation for the error terms. Noticing that the error terms all depend on the quantity $\frac{\gamma_k}{1-e^{-2(T-t_k)}}$ which is of order

$$\frac{\gamma_k}{1-e^{-2(T-t_k)}} = \begin{cases} \Theta(\gamma_k), & \text{if } T-t_k > 1, \\ \Theta(\dfrac{\gamma_k}{T-t_k}), & \text{if } T-t_k < 1. \end{cases}$$

Therefore

$$\sum_{k=1}^{N-1} \frac{(d+\mathrm{m}_2^2)\gamma_k\gamma_{k-1}^2}{(1-e^{-2(T-t_k)})(1-e^{-2(T-t_{k-1})})^2} + \sum_{k=0}^{N-1} \frac{d\gamma_k^2}{(1-e^{-2(T-t_k)})^2}$$

$$= \Theta\left( \sum_{1<T-t_k<T} (d+\mathrm{m}_2^2)\gamma_k^3 + d\gamma_k^2 + \sum_{\delta<T-t_k<1} \frac{(d+\mathrm{m}_2^2)\gamma_k^3}{(T-t_k)^3} + \frac{d\gamma_k^2}{(T-t_k)^2} \right)$$

Noticing that $x \mapsto x^2$ and $x \mapsto x^3$ are both convex functions on the domain $x \in (0,\infty)$. Since $\sum_{1<T-t_k<T} \gamma_k = T-1$ is fixed, according to Jensen's inequality, $\sum_{1<T-t_k<T} \gamma_k^2$ and $\sum_{1<T-t_k<T} \gamma_k^3$ reach their minimum when $\gamma_k$ are constant-valued for all $k$ such that $T-t_k > 1$. Similarly, let $\beta_k = \ln\left(\frac{T-t_k}{T-t_{k+1}}\right) \in (0,\infty)$. Then $\frac{\gamma_k}{T-t_k} = 1-e^{-\beta_k}$ and $\sum_{\delta<T-t_k<1} \beta_k = \ln(1/\delta)$ is fixed. Since $x \mapsto (1-e^{-x})^2$ and $x \mapsto (1-e^{-x})^3$ are both convex functions on the domain $x \in (0,\infty)$, according the Jensen's inequality, $\sum_{\delta<T-t_k<1} \frac{\gamma_k^2}{(T-t_k)^2}$ and $\sum_{\delta<T-t_k<1} \frac{\gamma_k^3}{(T-t_k)^3}$ reach their minimum when $\beta_k$ are constant-valued for all $k$ such that $T-t_k < 1$.

**Comparison to convergence results in denoising diffusion models.** (9) in Theorem 1 bounds the error of DDMC by I, II, III, which reflect the initialization error, the discretization error and the score estimation error, respectively. Assuming the score estimation error is small, (9) reduces to the same type of results that study the error bound for the denoising diffusion models (Algorithm 1). In [5], the discretization error is proved to be of order $\mathcal{O}(d)$ assuming the score function is smooth along the trajectory and a constant step-size. In [4], they get rid of the trajectory smoothness assumption and prove a $\mathcal{O}(d^2)$ discretization error bound with early-stopping. In [3], the discretization error bound is improved to $\mathcal{O}(d)$ with early-stopping and exponential-decay step-size, and without the trajectory smoothness assumption. Compared to these works, our result in Theorem 1 also implies a $\mathcal{O}(d)$ discretization error without the trajectory smoothness assumption and it applies to any choice of step-size with early stopping, as we discussed above. As shown in [3], the $\mathcal{O}(d)$ is the optimal for the discretization error. Therefore, our results indicates that with early-stopping, the denoising diffusion model achieves the optimal linear dimension dependent error bound.

## C.6 Proofs of Section 3.3

To prove the query complexity of ZOD-MC, we first look at query complexity of Algorithm 3.

***Query complexity of Algorithm 3.*** The query complexity of Algorithm 3 is essentially the number of proposals we need so that $n(t)$ of them can be accepted. Intuitively, to get one sample accepted, the number of proposals we need is geometrically distributed with certain acceptance probability [9]. We state this formally in the following proposition, for which it suffices to assume a relaxation of the commonly used gradient-Lipschitz condition on the potential.

**Proposition C.3.** *Under Assumption 3.2, the expected number of proposals for obtaining $n(t)$ many exact samples from $p_{0|t}(\cdot|x)$ defined in Lemma 1 via Algorithm 3, is*

$$N(t) = n(t)\left( \left(L(e^{2t}-1)+1\right)^{\frac{d}{2}} \exp\left(\frac{1}{2}\frac{\|Lx^*-e^tx\|^2}{L(e^{2t}-1)+1}\right) \right).$$

**Remark 5.** *Our complexity bound in Proposition C.3 exponentially depends on the dimension. This is due the curse of dimensionality phenomenon in the rejection sampling: the acceptance rate and algorithm efficiency decreases significantly when the dimension increases.*

*Proof of Proposition C.3.* For each $t \in [0,T]$, the expected number of iterations in the rejection sampling to get one accepted sample is

$$M(t) = \left( \int_{\mathbb{R}^d} e^{-V(z)+V(x^*)}\mathcal{N}\left(z; e^tx, (e^{2t}-1)I_d\right)\mathrm{d}z \right)^{-1}$$

$$\leq \left( \int_{\mathbb{R}^d} \exp\left(-\frac{L}{2}\|z-x^*\|^2\right)\mathcal{N}\left(z; e^tx, (e^{2t}-1)I_d\right)\mathrm{d}z \right)^{-1}$$

$$= \left(L(e^{2t}-1)+1\right)^{\frac{d}{2}} \exp\left(\frac{1}{2}\frac{\|Lx^*-e^tx\|^2}{L(e^{2t}-1)+1}\right).$$

To get $n(t)$ many samples, the expected number of iterations we need is $N(t) = n(t)M(t)$. $\qquad\square$

**Query complexity of ZOD-MC.** The query complexity of ZOD-MC, denoted as $\tilde{N}$, is essentially the sum of query complexities in Proposition C.3 over the discretized time points, i.e. $\tilde{N} = \sum_{k=0}^{N-1} N(T - t_k)$.

*Proof of Corollary 3.1.* First, for $\delta = \Theta\big(\min(\frac{\varepsilon_{W_2}^2}{d}, \frac{\varepsilon_{W_2}}{m_2})\big)$, it follows from Proposition C.1 that $W_2(p, p_\delta) \leq \varepsilon_{W_2}$.

Next, under the exponential-decay step size, according to Theorem 1, if we set $n(T - t_k) = \gamma_k^{-1} e^{-2(T-t_k)}$, then

$$\mathrm{KL}(p_\delta | q_{t_N}) \lesssim (d + m_2^2) e^{-2T} + \frac{(d + m_2^2)}{N^2}\big(T + \ln(\tfrac{1}{\delta})\big)^3 + \frac{d}{N}\big(T + \ln(\tfrac{1}{\delta})\big)^2.$$

By choosing $T = \frac{1}{2}\ln(\frac{d + m_2^2}{\varepsilon_{\mathrm{KL}}})$, $N = \Theta\big(\max(\frac{(d + m_2^2)^{\frac{1}{2}}(T + \ln(\delta^{-1}))^{\frac{3}{2}}}{\varepsilon_{\mathrm{KL}}^{\frac{1}{2}}}, \frac{d(T + \ln(\delta^{-1}))^2}{\varepsilon_{\mathrm{KL}}})\big)$ and $\kappa = \Theta\big(\frac{T + \ln(\delta^{-1})}{N}\big)$, we have $\mathrm{KL}(p_\delta | q_{t_N}) \lesssim \varepsilon_{\mathrm{KL}}$. Last, it follows from Proposition C.3 that

$$\tilde{N} \leq \sum_{k=0}^{N-1} \gamma_k^{-1} e^{-2(T-t_k)}\left(\big(L(e^{2(T-t_k)} - 1) + 1\big)^{\frac{d}{2}} \exp\big(\frac{1}{2}\frac{\|Lx^* - e^{T-t_k}x_k\|^2}{L(e^{2(T-t_k)} - 1) + 1}\big)\right).$$

By plugging in $\delta, T, t_k$ and $N$, (10) is proved.

$\square$

## C.7 Side Lemmas

**Lemma 2.** *Let $\{\bar{X}_t\}_{0 \leq t \leq T}$ be the solution to (2). Then for any $0 \leq k \leq N - 1$, we have*

$$\int_{t_k}^{t_{k+1}} \mathbb{E}\big[\big\|\nabla \ln p_{T-t}(\bar{X}_t) - \nabla \ln p_{T-t_k}(\bar{X}_{t_k})\big\|^2\big]\mathrm{d}t$$

$$\lesssim \frac{d\gamma_k^2}{(1 - e^{-2(T-t_k)})^2} + \frac{e^{-2(T-t_k)}\gamma_k}{(1 - e^{-2(T-t_k)})^2}\left(\mathbb{E}\big[trace\big(\Sigma_{T-t_k}(X_{T-t_k})\big)\big] - \mathbb{E}\big[trace\big(\Sigma_{T-t_{k+1}}(X_{T-t_{k+1}})\big)\big]\right),$$

*where $\{\Sigma_t(X_t)\}_{0 \leq t \leq T}$ is defined in Proposition C.2.*

*Proof of Lemma 2.* For fixed $s$, consider the process $\{\nabla \ln p_{T-t}(\bar{X}_t)\}_{0 \leq t \leq T}$, denoted as $\{L_t\}_{0 \leq t \leq T}$, and a function $E_{s,t} := \mathbb{E}[\|L_t - L_s\|^2]$. It is shown by Itô's formula in [3, Lemma 3] that

$$\mathrm{d}L_t = -L_t \mathrm{d}t + \sqrt{2}\nabla^2 \ln q_{T-t}(\bar{X}_t)\mathrm{d}\bar{B}_t, \tag{32}$$

and as a result, (32) implies that

$$\mathrm{d}E_{s,t} = -2E_{s,t}\mathrm{d}t - 2\mathbb{E}\big[\langle L_t - L_s, L_s \rangle\big]\mathrm{d}t + 2\mathbb{E}\big[\big\|\nabla^2 \ln p_{T-t}(\bar{X}_t)\big\|_F^2\big]\mathrm{d}t, \tag{33}$$

Apply (32) and Itô's formula again, we have

$$\mathrm{d}\mathbb{E}\big[\langle L_t, L_s \rangle\big] = -\mathbb{E}\big[\langle L_t, L_s \rangle\big]\mathrm{d}t \quad \Longrightarrow \quad \mathbb{E}\big[\langle L_t, L_s \rangle\big] = e^{-(t-s)}\mathbb{E}\big[\|L_s\|^2\big].$$

Therefore (33) can be rewritten as

$$\frac{\mathrm{d}}{\mathrm{d}t}E_{s,t} = -2E_{s,t} + 2(1 - e^{-(t-s)})\mathbb{E}\big[\|L_s\|^2\big] + 2\mathbb{E}\big[\big\|\nabla^2 \ln p_{T-t}(\bar{X}_t)\big\|_F^2\big]. \tag{34}$$

Let $\{X_t\}_{0 \leq t \leq T}$ be the solution of (1). Since $\bar{X}_t = X_{T-t}$ in distribution for all $t \in [0, T]$, $\mathbb{E}\big[\|L_s\|^2\big]$ and $\mathbb{E}\big[\big\|\nabla^2 \ln p_{T-t}(\bar{X}_t)\big\|_F^2\big]$ can both be represented by the covariance matrix defined in Proposition C.2. It is proved in [3, Lemma 6] that

$$\mathbb{E}\big[\|L_s\|^2\big] = \mathbb{E}\big[\big\|\nabla \ln p_{T-s}(\bar{X}_s)\big\|^2\big]$$

$$= \frac{d}{1 - e^{-2(T-s)}} - \frac{e^{-2(T-s)}}{(1 - e^{-2(T-s)})^2}\mathbb{E}\big[\mathrm{trace}\big(\Sigma_{T-s}(X_{T-s})\big)\big] \tag{35}$$

and

$$\mathbb{E}\big[\big\|\nabla^2 \ln p_{T-t}(\bar{X}_t)\big\|_F^2\big] = \frac{d}{(1-e^{-2(T-t)})^2} - \frac{e^{-2(T-t)}}{2(1-e^{-2(T-t)})^2}\frac{\mathrm{d}}{\mathrm{d}t}\bigg(\mathbb{E}\big[\mathrm{trace}\big(\Sigma_{T-t}(X_{T-t})\big)\big]\bigg)$$
$$- \frac{2e^{-2(T-t)}}{(1-e^{-2(T-t)})^3}\mathbb{E}\big[\mathrm{trace}\big(\Sigma_{T-t}(X_{T-t})\big)\big]. \tag{36}$$

Now we choose $s = t_k$ in (34) and integrate from $t_k$ to $t$. According to (35) and (36), we have

$$\frac{1}{2}e^{2t}E_{t_k,t_{k+1}} = d\int_{t_k}^t \frac{e^{2u}-e^{u+t_k}}{1-e^{-2(T-t_k)}} + \frac{e^{2u}}{(1-e^{-2(T-u)})^2}\mathrm{d}u$$

$$- \frac{e^{-2(T-t_k)}}{(1-e^{-2(T-t_k)})^2}\mathbb{E}\big[\mathrm{trace}\big(\Sigma_{T-t_k}(X_{T-t_k})\big)\big]\int_{t_k}^t e^{2u}-e^{u+t_k}\mathrm{d}u$$

$$- \int_{t_k}^t \frac{e^{-2(T-u)+2u}}{2(1-e^{-2(T-u)})^2}\mathrm{d}\mathbb{E}\big[\mathrm{trace}\big(\Sigma_{T-u}(X_{T-u})\big)\big]$$

$$- 2\int_{t_k}^t \frac{e^{-2(T-u)+2u}}{(1-e^{-2(T-u)})^3}\mathbb{E}\big[\mathrm{trace}\big(\Sigma_{T-u}(X_{T-u})\big)\big]\mathrm{d}u$$

$$= \frac{d}{2}\bigg(\frac{e^{2t}+e^{2t_k}-2e^{t_k+t}}{1-e^{-2(T-t_k)}} + \frac{e^{2t}-e^{2t_k}}{(1-e^{-2(T-t_k)})(1-e^{-2(T-t)})}\bigg)$$

$$- \frac{e^{-2(T-t_k)}}{(1-e^{-2(T-t_k)})^2}\mathbb{E}\big[\mathrm{trace}\big(\Sigma_{T-t_k}(X_{T-t_k})\big)\big]\big(e^{2t}+e^{2t_k}-2e^{t_k+t}\big),$$

$$+ \frac{e^{-2(T-t_k)+2t_k}}{(1-e^{-2(T-t_k)})^2}\mathbb{E}\big[\mathrm{trace}\big(\Sigma_{T-t_k}(X_{T-t_k})\big)\big] - \frac{e^{-2(T-t)+2t}}{(1-e^{-2(T-t)})^2}\mathbb{E}\big[\mathrm{trace}\big(\Sigma_{T-t}(X_{T-t})\big)\big]$$

$$- \int_{t_k}^t \frac{e^{-2(T-u)+2u}}{(1-e^{-2(T-u)})^2}\mathbb{E}\big[\mathrm{trace}\big(\Sigma_{T-u}(X_{T-u})\big)\big]\mathrm{d}u,$$

where the last identity follows from integration by parts. According to Proposition C.2, $t \mapsto \mathbb{E}\big[\mathrm{trace}\big(\Sigma_{T-t}(X_{T-t})\big)\big]$ is positive and decreasing. Therefore, we have for all $t \in [t_k, t_{k+1}]$,

$$E_{t_k,t} \leq d\bigg(\frac{1+e^{-2(t-t_k)}-2e^{-(t-t_k)}}{1-e^{-2(T-t_k)}} + \frac{1-e^{-2(t-t_k)}}{(1-e^{-2(T-t_k)})(1-e^{-2(T-t)})}\bigg)$$

$$+ 2\frac{e^{-2(T-t_k)}}{(1-e^{-2(T-t_k)})^2}\big(2e^{-(t-t_k)}-1\big)\mathbb{E}\big[\mathrm{trace}\big(\Sigma_{T-t_k}(X_{T-t_k})\big)\big]$$

$$- \bigg(\frac{2e^{-2(T-t)}}{(1-e^{-2(T-t)})^2} + 2\int_{t_k}^t \frac{e^{-2(T-t_k)-2(t-u)}}{(1-e^{-2(T-t_k)})^2}\mathrm{d}u\bigg)\mathbb{E}\big[\mathrm{trace}\big(\Sigma_{T-t}(X_{T-t})\big)\big].$$

Integrate again from $t = t_k$ to $t = t_{k+1}$, we get

$$\int_{t_k}^{t_{k+1}} E_{t_k,t}\mathrm{d}t$$

$$\leq \frac{d\gamma_k^3}{1-e^{-2(T-t_k)}} + \frac{2d\gamma_k^2}{(1-e^{-2(T-t_k)})^2} + \frac{2e^{-2(T-t_k)}}{(1-e^{-2(T-t_k)})^2}\big(2-2e^{-\gamma_k}-\gamma_k\big)\mathbb{E}\big[\mathrm{trace}\big(\Sigma_{T-t_k}(X_{T-t_k})\big)\big]$$

$$- \frac{e^{-2(T-t_k)}}{(1-e^{-2(T-t_k)})^2}\big(3\gamma_k+\frac{1}{2}e^{-\gamma_k}-\frac{1}{2}\big)\mathbb{E}\big[\mathrm{trace}\big(\Sigma_{T-t_{k+1}}(X_{T-t_{k+1}})\big)\big]$$

$$\leq \frac{3d\gamma_k^2}{(1-e^{-2(T-t_k)})^2} + \frac{2e^{-2(T-t_k)}\gamma_k}{(1-e^{-2(T-t_k)})^2}\big(\mathbb{E}\big[\mathrm{trace}\big(\Sigma_{T-t_k}(X_{T-t_k})\big)\big] - \mathbb{E}\big[\mathrm{trace}\big(\Sigma_{T-t_{k+1}}(X_{T-t_{k+1}})\big)\big]\big).$$

$\square$

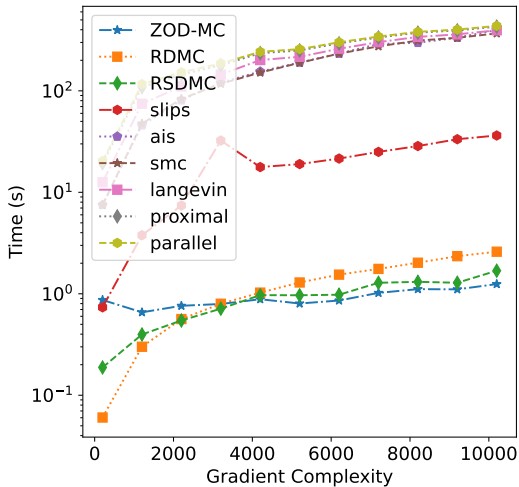

Figure 7: Wall clock of different methods as a function of oracle complexity for the 2D GMM in the main paper

# D   More Experiments

## D.1   Samples from 2D GMM at different Oracle Complexities

We sample from a Gaussian Mixture model with $4$ modes, the following summarizes the parameters of the GMM.

$$\text{Weights, } \boldsymbol{w} : \begin{bmatrix} 0.1 & 0.2 & 0.3 & 0.4 \end{bmatrix},$$

$$\text{Means, } \boldsymbol{\mu_k} : \begin{bmatrix} 0 \\ 0 \end{bmatrix}, \begin{bmatrix} 0 \\ 11 \end{bmatrix}, \begin{bmatrix} 9 \\ 9 \end{bmatrix}, \begin{bmatrix} 11 \\ 0 \end{bmatrix},$$

$$\text{Covariances, } \boldsymbol{\Sigma_k} : \begin{bmatrix} 1 & 0.5 \\ 0.5 & 1 \end{bmatrix}, \begin{bmatrix} 0.3 & -0.2 \\ -0.2 & 0.3 \end{bmatrix}, \begin{bmatrix} 1 & 0.3 \\ 0.3 & 1 \end{bmatrix}, \begin{bmatrix} 1.2 & -1 \\ -1 & 1.2 \end{bmatrix}.$$

We display the generated samples at different oracle complexities in Figures 8, 9, 10. Notice that the mode located at the origin holds less weight, and as the oracle complexity increases our method becomes better at sampling from other modes, as opposed to the corresponding baselines.

We detail the hyperparameters in Table 2 and the wall clock time of different methods in Figure 15.

| Method | T | N | $\delta$ | Step Size | N-MCMC | Num Steps | N-Chains |
|---|---|---|---|---|---|---|---|
| ZOD-MC | 2 | 25 | $5e\text{-}3$ | - | $K$ | - | - |
| RDMC | 2 | 25 | $5e\text{-}2$ | 0.01 | 1000 | $K/1000$ | - |
| RSDMC | 2 | 25 | $5e\text{-}2$ | 0.01 | $K^{1/4}$ | $K^{1/4}$ | - |
| SLIPS | 1 | 25 | $6.62e\text{-}3$ | Adaptive | 1000 | $K/1000$ | - |
| AIS | - | - | - | Adaptive | - | M | 512 |
| SMC | - | - | - | Adaptive | - | M | 512 |
| Langevin | - | - | - | 0.01 | - | M | - |
| Proximal | - | - | - | $1/5$ | - | M | - |
| Parallel | - | - | - | 0.01 | - | M | 512 |

Table 2: Hyperparameters for Various Methods for the 2D GMM experiment. $K$ means the current oracle complexity and $M$ refers to a matched oracle complexity. For RSDMC we used 2 recursions per score evaluation.

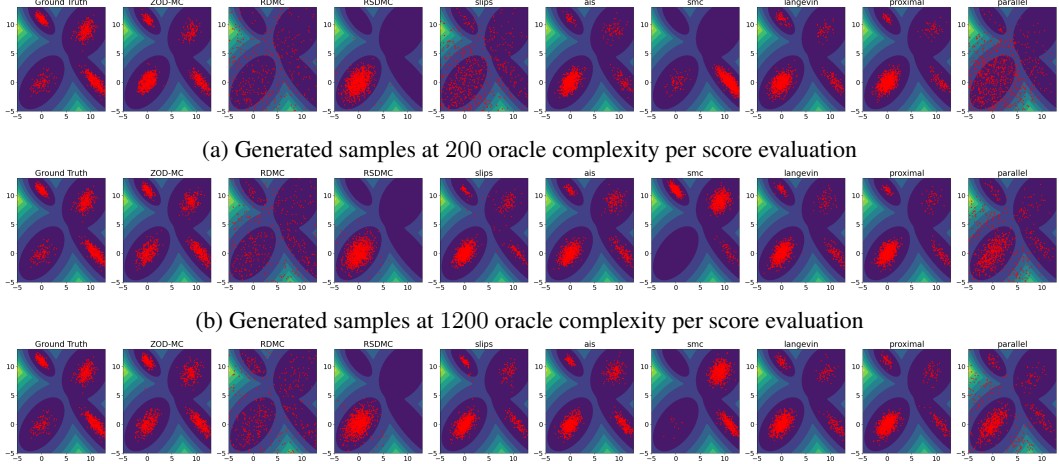

(a) Generated samples at 200 oracle complexity per score evaluation

(b) Generated samples at 1200 oracle complexity per score evaluation

(c) Generated samples at 2200 oracle complexity per score evaluation

Figure 8: Generated Samples for GMM at different oracle complexity

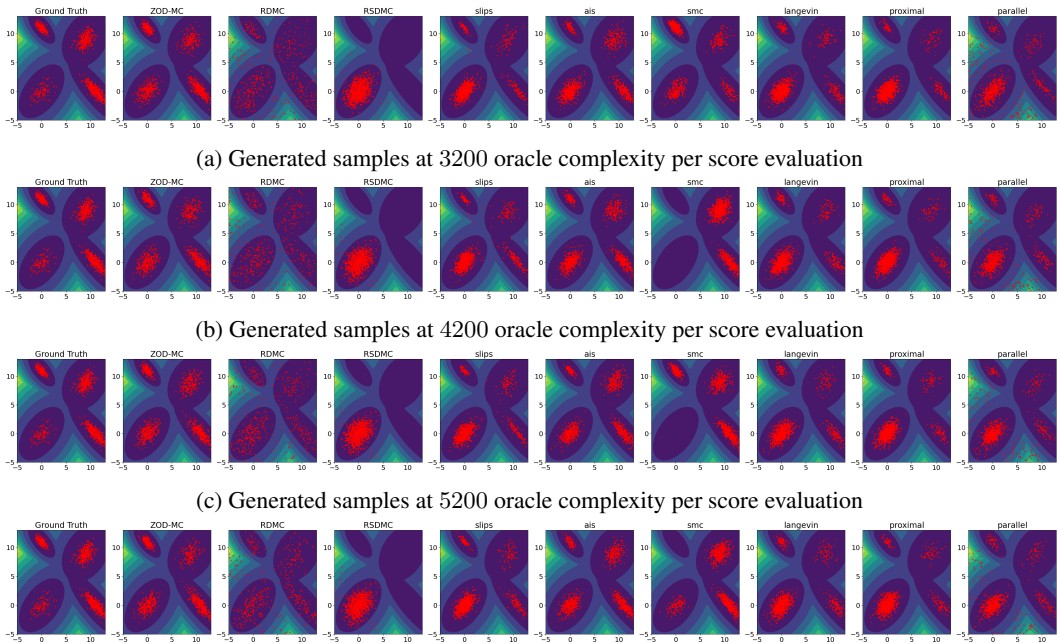

(a) Generated samples at 3200 oracle complexity per score evaluation

(b) Generated samples at 4200 oracle complexity per score evaluation

(c) Generated samples at 5200 oracle complexity per score evaluation

(d) Generated samples at 6200 oracle complexity per score evaluation

Figure 9: Generated Samples for GMM at different oracle complexity

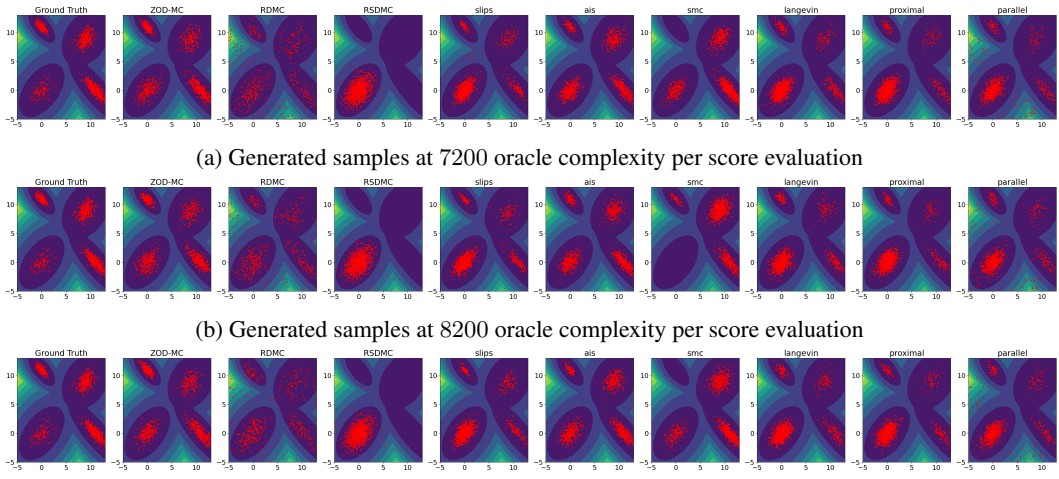

(a) Generated samples at 7200 oracle complexity per score evaluation

(b) Generated samples at 8200 oracle complexity per score evaluation

(c) Generated samples at 9200 oracle complexity per score evaluation

Figure 10: Generated Samples for GMM at different oracle complexity

## D.2 Samples from Discontinuous 2D GMM at different Oracle Complexities

We display the generated samples at different oracle complexities in Figures 11 ,12, 13. At the end we show the $W_2$ and MMD for this example in Figure 14.

We detail the hyperparameters in Table 3 .

| Method | T | N | $\delta$ | Step Size | N-MCMC | Num Steps | N-Chains |
|--------|---|---|----------|-----------|--------|-----------|----------|
| ZOD-MC | 2 | 25 | $5e$-3 | - | $K$ | - | - |
| RDMC | 2 | 25 | $5e$-2 | 0.01 | 1000 | $K/1000$ | - |
| RSDMC | 2 | 25 | $5e$-2 | 0.01 | $K^{1/4}$ | $K^{1/4}$ | |
| SLIPS | 1 | 25 | $6.62e$-3 | Adaptive | 1000 | $K/1000$ | - |
| AIS | - | - | - | Adaptive | - | M | 512 |
| SMC | - | - | - | Adaptive | - | M | 512 |
| Langevin | - | - | - | 0.01 | - | M | - |
| Proximal | - | - | - | 1/5 | - | M | - |
| Parallel | - | - | - | 0.01 | - | M | 512 |

Table 3: Hyperparameters for Various Methods for the 2D Discontinuous GMM experiment. $K$ means the current oracle complexity and $M$ refers to a matched oracle complexity. For RSDMC we used 2 recursions per score evaluation.

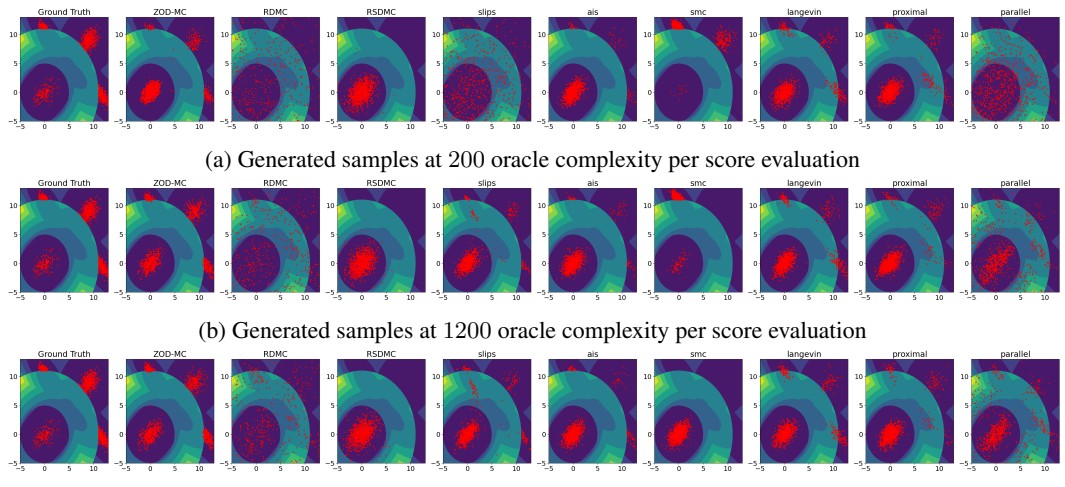

(a) Generated samples at 200 oracle complexity per score evaluation

(b) Generated samples at 1200 oracle complexity per score evaluation

(c) Generated samples at 2200 oracle complexity per score evaluation

Figure 11: Generated Samples for discontinuous GMM at different oracle complexity

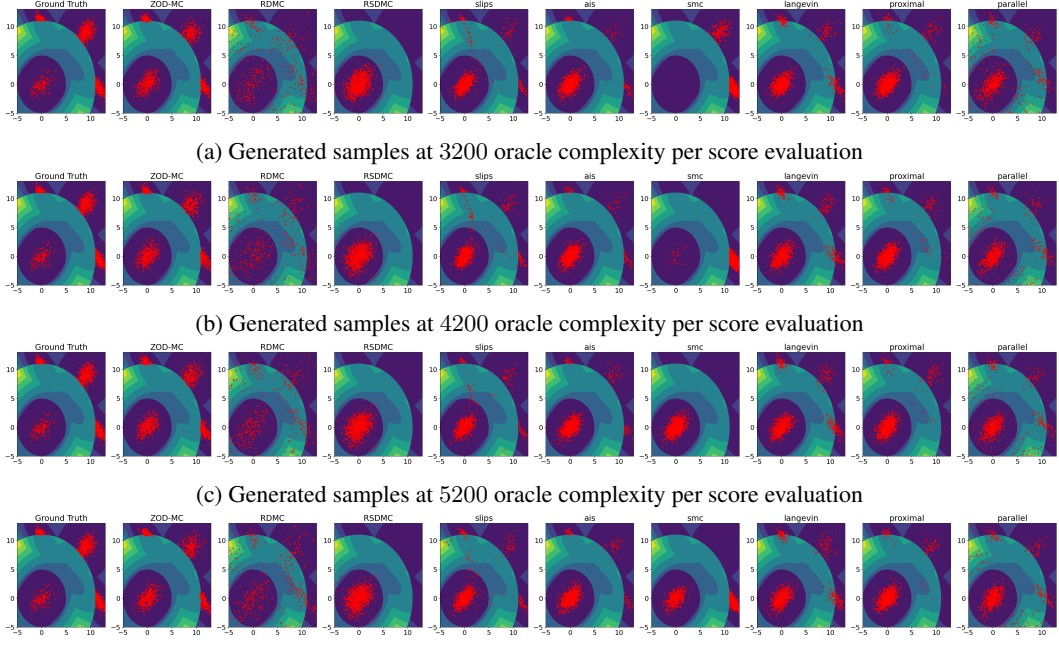

(a) Generated samples at 3200 oracle complexity per score evaluation

(b) Generated samples at 4200 oracle complexity per score evaluation

(c) Generated samples at 5200 oracle complexity per score evaluation

(d) Generated samples at 6200 oracle complexity per score evaluation

Figure 12: Generated Samples for discontinuous GMM at different oracle complexity

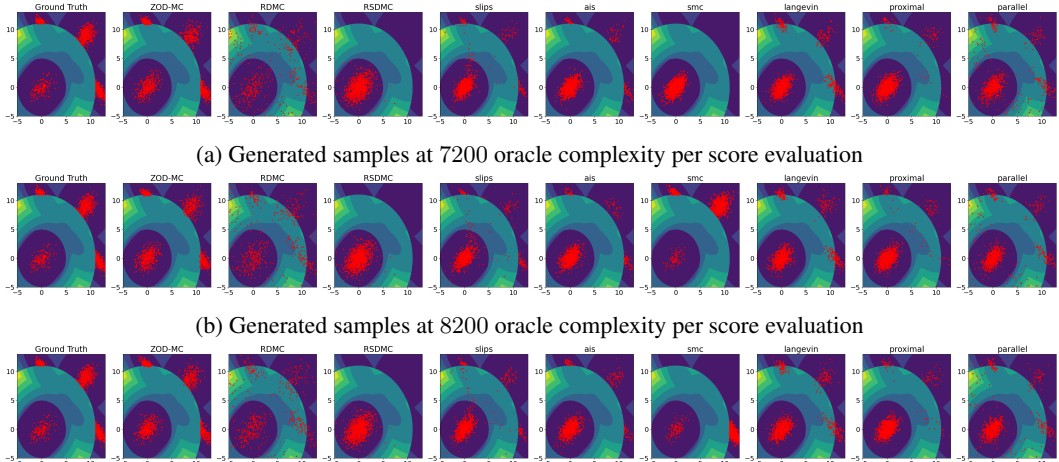

(a) Generated samples at 7200 oracle complexity per score evaluation

(b) Generated samples at 8200 oracle complexity per score evaluation

(c) Generated samples at 9200 oracle complexity per score evaluation

Figure 13: Generated Samples for discontinuous GMM at different oracle complexity

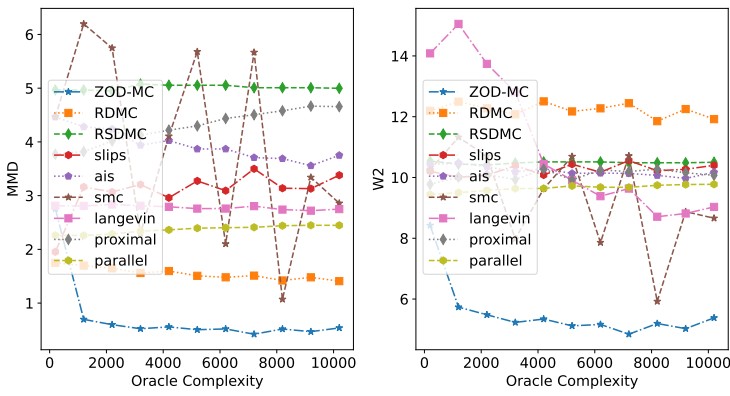

Figure 14: $W_2$ and MMD at different oracle complexities for discontinuous potential

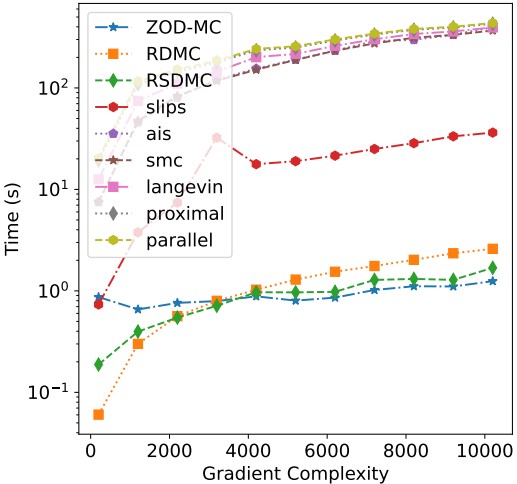

Figure 15: Wall clock of different methods as a function of oracle complexity

### D.3 Samples from different radius

We display the generated samples at different radius in Figures 16, 18 and the hyperparameters in Table 4.

| Method | T | N | $\delta$ | Step Size | N-MCMC | Num Steps | N-Chains |
|--------|---|---|----------|-----------|--------|-----------|----------|
| ZOD-MC | 10 | 50 | $5e$-3 | - | $K$ | - | - |
| RDMC | 2 | 50 | $5e$-2 | 0.01 | 1000 | $K/1000$ | - |
| RSDMC | 2 | 50 | $5e$-2 | 0.01 | $K^{1/4}$ | $K^{1/4}$ | - |
| SLIPS | 1 | 50 | $6.62e$-3 | Adaptive | 1000 | $K/1000$ | - |
| AIS | - | - | - | Adaptive | - | M | 512 |
| SMC | - | - | - | Adaptive | - | M | 512 |
| Langevin | - | - | - | 0.01 | - | M | - |
| Proximal | - | - | - | $1/40$ | - | M | - |
| Parallel | - | - | - | 0.01 | - | M | 512 |

Table 4: Hyperparameters for Various Methods for the 2D GMM experiment. $K$ means the current oracle complexity and $M$ refers to a matched oracle complexity.

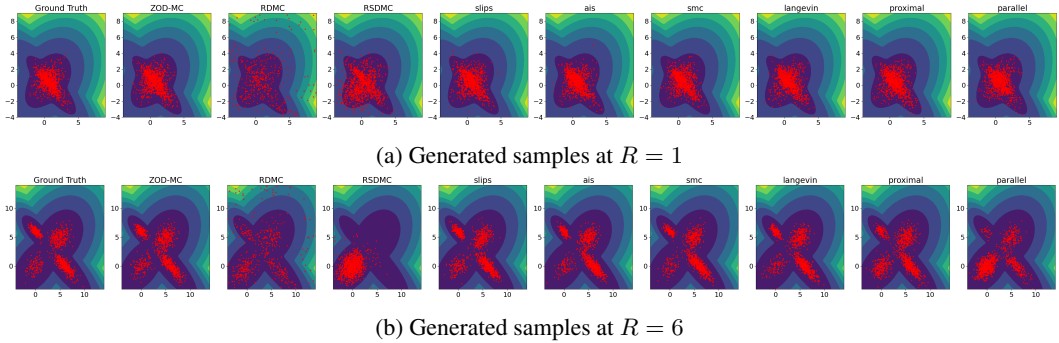

(a) Generated samples at $R = 1$

(b) Generated samples at $R = 6$

Figure 16: Generated samples at different radius

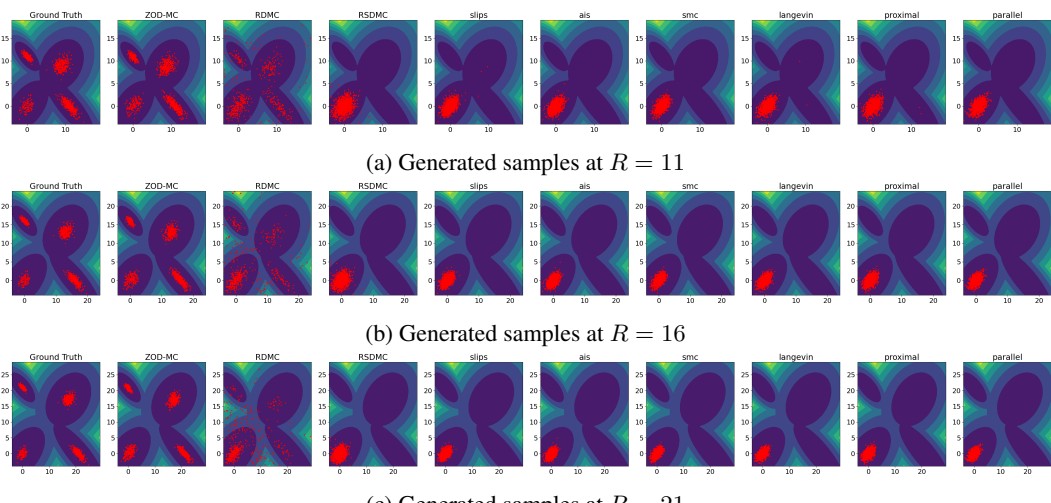

(a) Generated samples at $R = 11$

(b) Generated samples at $R = 16$

(c) Generated samples at $R = 21$

Figure 17: Generated samples at different radius

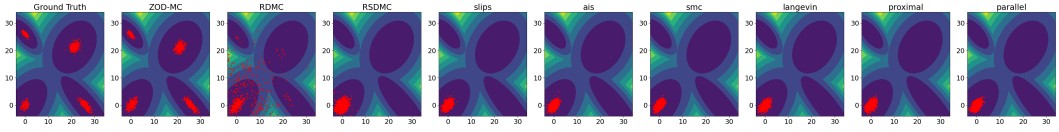

(a) Generated samples at $R = 26$

Figure 18: Generated samples at different radius

## D.4 Higher Dimensional Examples

**Score Error Approximation Details.** We use the following 5d Gaussian mixture to measure the error of the score approximation:

$$\text{Coefficients, } \boldsymbol{w} : \begin{bmatrix} 0.25 & 0.5 & 0.25 \end{bmatrix},$$

$$\text{Means, } \boldsymbol{\mu_k} : \begin{bmatrix} -4 \\ -4 \\ -3 \\ -4 \\ -4 \end{bmatrix}, \begin{bmatrix} 4 \\ 3 \\ 4 \\ 2 \\ 4 \end{bmatrix}, \begin{bmatrix} -4 \\ -2 \\ -4 \\ 4 \\ -1 \end{bmatrix},$$

$$\text{Variances, } \boldsymbol{\Sigma_k} : \begin{bmatrix} 3 & 2 & 0 & 0 & 0 \\ 2 & 3 & 0 & 0 & 0 \\ 0 & 0 & 4 & 2 & 0 \\ 0 & 0 & 2 & 4 & 0 \\ 0 & 0 & 0 & 0 & 1 \end{bmatrix}, \begin{bmatrix} 9 & 0 & 7 & 0 & 0 \\ 0 & 1 & 0 & 0.4 & 0 \\ 7 & 0 & 9 & 0 & 0 \\ 0 & 0.4 & 0 & 1 & 0 \\ 0 & 0 & 0 & 0 & 1 \end{bmatrix}, \begin{bmatrix} 1 & 0.4 & 0 & 0 & 0 \\ 0.4 & 1 & 0 & 0 & 0 \\ 0 & 0 & 4 & 3 & 0 \\ 0 & 0 & 3 & 4 & 0 \\ 0 & 0 & 0 & 0 & 1 \end{bmatrix}.$$

**Randomized Gaussian Mixtures** To generate the results in Figure 1b we proceed as follows. For a given dimension we take:

$$\mu = \frac{z}{\|z\|} \cdot 12$$

Where $z \sim U[0,1]^d$, additionally we sample $\sigma^2 \sim U[.3, 1.3]$. We then consider the Gaussian target $\mathcal{N}(\mu, \sigma^2 I)$. We repeat this 5 times and create a Gaussian mixture with equally weighted modes. We plot the the 2d marginals of the target distribution as long as the generated samples.

### D.5 Muller Brown Potential Details

The potential is given by $V(x, y) = \beta \cdot (V_m(x, y) + V_q(x, y))$

$$
\begin{aligned}
V_m(x, y) = &-170 \exp\left(-6.5(x+0.5)^2 + 11(x+0.5)(y-1.5) - 6.5(y-1.5)^2\right) \\
&- 100 \exp\left(-x^2 - 10(y-0.5)^2\right) + 15 \exp\left(0.7(x+1)^2 + 0.6(x+1)(y-1) + 0.7(y-1)^2\right) \\
&- 200 \exp\left(-(x-1)^2 - 10y^2\right),
\end{aligned}
$$

where $V_m$ corresponds to the original Müller Brown and $V_q(x, y) = 35.0136(x-x_c^*)^2 + 59.8399(y - y_c^*)^2$, with $(x_c^*, y_c^*)$ is approximately the minimizer at the center of the middle potential well, and $V_q$ is a correction introduced so that the depths of all three wells are.

### D.6 Score error at $t = T$

One natural concern is that the sampling problem at $t = T$ could be nearly as hard as sampling from the target distribution. Therefore, only a small number of samples could be accepted and the score error would be high. We display the score error at $t = T$ to show that this is not necessarily the case.

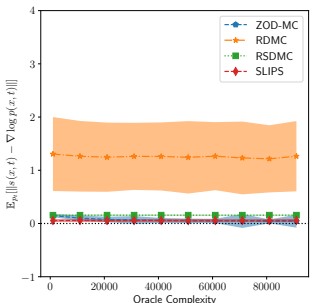 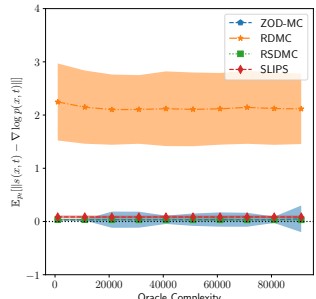

(a) Score error at $t = T = 4$. for different diffusion samplers, we used $t \approx 1$ for SLIPS. Results are for the 2d GMM in the main paper

(b) Score error at $t = T = 4$. for different diffusion samplers, we used $t \approx 1$ for SLIPS. Results are for the 5d GMM in the main paper

Figure 19: Score error at $t = T$ for different target distributions

### D.7 Further details on number of accepted samples

We present the number of accepted samples from our rejection sampler as a function of time. Specifically we consider 1000 trajectories of the diffusion and for every intermediate step we sample $10K$ samples. We present the number of accepted samples in Table 5.

| $t_0$ | 5.00 | 4.28 | 3.56 | 2.84 | 2.13 | 1.41 | 0.69 | 0.30 | 0.13 | 0.01 |
|---|---|---|---|---|---|---|---|---|---|---|
| GMM | 1.58 | 1.39 | 3.70 | 14.03 | 48.27 | 129.05 | 308.23 | 786.74 | 1299.30 | 2251.65 |
| Mueller | 1.01 | 1.20 | 2.62 | 9.79 | 37.25 | 145.42 | 447.96 | 945.72 | 1598.99 | 3129.97 |

Table 5: Comparison of GMM and Mueller values across different $t$ values

