# OpenReview forum: "Zeroth-Order Sampling Methods for Non-Log-Concave Distributions: Alleviating Metastability by Denoising Diffusion"
_NeurIPS.cc/2024/Conference — NeurIPS 2024 poster_

### Official Review · Reviewer_HU9Z · 2024-07-06

**Soundness:** 3
**Presentation:** 3
**Contribution:** 3
**Rating:** 7
**Confidence:** 3

**Summary:**

The authors investigate a popular reverse-diffusion sampling process, when the scores are estimated not from target samples but using the target's unnormalized density, which is the setup in energy-based modelling. Specifically, the authors consider a recent Monte-Carlo estimator of the scores which requires sampling from a possibly multimodal distribution (product of target and Gaussian densities). To do this, the authors use rejection sampling and obtain an upper bound on the convergence of their method. They also empirically validate their method in low dimensions with thorough experiments that monitor and benchmark against other methods:

- **convergence** in MMD, W2, and generic statistics vs. **cost** in dimension or target queries) in Figure 1
- quantifying **mode coverage** in Figure 4

as well as other visual diagnostics of convergence.

**Strengths:**

The paper is clearly written and the results are interesting. Their experiments are thorough, I appreciate the use of the $W2$ in Fig 1.a. as it is sensitive to mode coverage which is less the case of some other sampling metrics.

**Weaknesses:**

The authors are clear about the limitations of their method, for example the complexity in Corollary 3.1. is exponential not only in the dimension $d$ but also in the smoothness $L$ of the target (this holds even in one dimension).

**Questions:**

-

**Limitations:**

Yes.

---

> ### Author Rebuttal · Authors · 2024-08-07
>
> We sincerely thank the reviewer for their valuable advice and comments and greatly appreciate the positive evaluation.
>
> >The authors are clear about the limitations of their method, for example the complexity in Corollary 3.1. is exponential not only in the dimension d but also in the smoothness L of the target (this holds even in one dimension).
>
> That is right. Just to further remark about this fact: For general target distributions as considered in our paper, it was shown in [1] that the diffusion posterior sampling is computationally intractable in the worse case, and that worse case is among the distributions considered in our paper. Therefore, we feel if the theoretical complexity needs to be improved, one possibility is to restrict the target distributions to have more structures, while possibly adapting our algorithm to these structures to get better smoothness/dimension-dependence.
>
> [1] Gupta, Shivam, et al. "Diffusion posterior sampling is computationally intractable." arXiv preprint arXiv:2402.12727 (2024).

---

> > ### Comment · Reviewer_HU9Z · 2024-08-11
> > **Answer to authors**
> >
> > I thank the authors for their answer.

---

### Official Review · Reviewer_Q2x6 · 2024-07-11

**Soundness:** 2
**Presentation:** 1
**Contribution:** 2
**Rating:** 5
**Confidence:** 5

**Summary:**

In this paper, the authors are interested in the problem of sampling from an arbitrary non-logconcave probability distribution (namely, multi-modal distribution) with only access to its unnormalized density. While most of popular sampling methods rely on queries of the score of the target, i.e. the gradient of the log-density (1st-order methods), the proposed approach is a 0th-order sampling method, as it relies on queries of the unnormalized target density itself. Inspired by the performance of denoising diffusion models for generative tasks, they propose a sampling algorithm, Zeroth-Order Diffusion Monte Carlo (ZOD-MC), which simulates a discretized and approximate version of the time-reversal of the standard Ornstein-Uhlenbeck diffusion process (the ideal scheme being known to converge to the target). In this algorithm, rejection-sampling is used to provide a Monte Carlo estimate of the intractable scores of the process marginals that appear in the recursion. This is made possible by the Tweedie's formula, that links the score of the marginals to an expectation over the posteriors of the model. As rejection sampling is known to suffer from the curse of dimensionality, the authors acknowledge that ZOD-MC cannot address sampling problems with relatively high dimensions ($d\geq10$). To support their algorithm, they provide a convergence analysis which requires a weak assumption on the target (bounded second order moment) and a control of the Monte Carlo estimation error for the posterior distributions at any time of the process. Under an extra-assumption on this control, they derive a readable oracle complexity of ZOD-MC that quantifies the number of queries of the target density in the whole sampling procedure. Finally, they conduct numerical experiments on low dimensional settings (mostly $d=2$), by considering multi-modal distributions which increasing higher energy barriers or with discontinuous potential. They compare ZOD-MC to recent approaches based on denoising approaches [1,2] and standard MCMC approaches such as Langevin or parallel tempering. In the considered settings, ZOD-MC exhibits the best performance at equivalent computational budget (namely, same oracle complexity)

[1] Reverse Diffusion Monte Carlo. Huang et al. 2023.

[2] Faster Sampling without Isoperimetry via Diffusion-based Monte Carlo. Huang et al. 2024

**Strengths:**

- The paper is didactic in the sense that the motivations and the bricks to build the algorithm ZOD-MC are exposed in a logical way.
- The authors pay attention to bring intuition on their theoretical results.
- The mathematical results are clear to understand, especially Theorem 1.
- Although the numerical experiments are conducted on low dimensional settings, they consider interesting cases, i.e., sampling from multi-modal distributions where the modes are increasingly further from each other and Gibbs distributions with discontinuous potential.

**Weaknesses:**

In my opinion, this paper suffers from 3 main weaknesses, which explains my score.

1. The comparison to the related work is incomplete. The authors do not cite the work [1] (although it was published approximately at the same time as [2]), where the authors propose a 1st-order sampling method, SLIPS, based on stochastic localization, that can be linked to a denoising diffusion model. In SLIPS, samples from the target distribution are obtained by following a diffusion process with intractable drift, which is nothing less than an expectation of a posterior distribution; in practice, the authors estimate this term by MCMC method (Langevin). In contrast to RDMC or ZOD-MC, SLIPS may exhibit multiple advantages:  (i) it can be applied to a flexible class of stochastic localization schemes, whereas RDMC and ZOD-MC only consider the standard Ornstein-Uhlenbeck diffusion process, (ii) it provides an exact finite-time setting for sampling (which is not the case for RDMC or ZOD-MC),  (iii) it is shown to scale well in dimension in their experiments (up to dimension 100), (iv) it has a small oracle complexity per score evaluation compared to the numerics presented here (32 vs 200 at least in the numerics of the current paper). Since the current approach has the same spirit as SLIPS (diffusion-based approach with MC estimation), it should be compared with it. Moreover, no numerical comparison with AIS or SMC (which are considered gold-standard to sample from low-dimensional multi-modal distributions) is provided. For me, this justifies why this paper does not meet the standards of NeurIPS.

2. In my opinion, the proposed approach misses a fundamental point, which lies in the MC estimation of the denoiser (i.e. the conditional expectation over the posterior distribution): it is linked to the ability to be able to provide an accurate MC estimate near at $t_0=0$, equivalently $t=T$, namely when $p_t$ is fully noised. When $t\approx T$, it is clear that the corresponding posterior distribution is approximately equal to the target distribution itself (since we have full noise, the quadratic term vanishes). Therefore, applying rejection sampling on the posterior distribution at the very beginning of the sampling process in ZOD-MC is equivalent to do rejection sampling on the target distribution itself ! Hence, in practice, **the whole diffusion-based approach presented in ZOD-MC to sample from the target distribution via an annealing scheme is actually as hard as directly sampling from the target distribution**, which hurts its usefulness. This remark is intimately linked to the notion of duality of log-concavity presented in [1], which explains that it is required to start the diffusion process further such that the posterior distribution is "smoother", and easier to sample from. On the theoretical side, I have a concerns that is linked to my point:  it seems to me that the authors make an assumption on the MC error $\delta(t)$ to estimate the denoiser  (which is not explicited in the main) in order to derive the total complexity cost (hidden in Section B.5 in the appendix); as said above, this control is actually crucial in practice and it is not straightforward to verify an upper bound on it.

3. The section on numerical experiments is incomplete for several reasons: no indication of tuning of the hyperparameters ($T$, $N$, step-size) either for ZOD-MC or other approaches, no ablation study on these hyperparameters to exhibit robustness, display of the estimation error of the true score for a large $t$ (where it becomes harder) and a variety of levels of oracle complexity, the plots in Figure 1 and Figure 4.a do not display any variance, missing related work as stated in my claim 1.

[1] Stochastic Localization via Iterative Posterior sampling. Grenioux et al. 2024

[2] Faster Sampling without Isoperimetry via Diffusion-based Monte Carlo. Huang et al. 2024

**Questions:**

- I have a question about Algorithm 1: to obtain the sampling recursion with the MC estimator, you first discretize the SDE with exponential integration and then replace the score by its MC estimator. Would it bring less discretization error to first replace the score in the SDE by the term involving the conditional expectation and then apply exponential integration upon it ? It seems to me that exponential integration would still be tractable in this case.
- To support my claim 2 in the 'Weaknesses' section, could the authors provide (i) the value of the average score error between the true score and the approximated score near $t=T$, while varying the number of queries to the target distribution (ie oracle complexity per score evaluation)  and (ii) results of rejection sampling directly applied on the target distribution with same oracle complexity as other methods ?

**Limitations:**

The main (and crucial) limitation of ZOD-MC is the fact that it can only be applied to small dimensional settings ($d\leq 5$), as rejection sampling is notoriously known to suffer from the curse of dimensionality (which is the reason why it is not a popular sampling method in practice). Although the authors acknowledge this limitation in the abstract, it is not well highlighted at all in the main of the paper: this limitation appears explicitly in Remark 5 in Section 3.3 in the Appendix, but not in the main paper. As suggested by NeurIPS guidelines, a section/paragraph 'Limitations' should be given in the main.

On the other hand, ZOD-MC requires to know the location of local minima of the energy landscape, i.e., the location of the modes of the target distribution. As indicated by the authors, this can be done by applying Newton's method on the target potential (see Remark 1) before or while sampling; however, this is a 2nd-order method, which violates the framework chosen by the authors (0-th order method). Although it is a common assumption to have access to the location of the modes in realistic settings (which turns out to be another challenge of sampling procedures), this requirement should be much more highlighted. Otherwise the title of the paper is misleading.

---

> ### Author Rebuttal · Authors · 2024-08-07
>
> We sincerely thank the reviewer for their valuable advice and comments.
>
> > The comparison to the related work is incomplete. The authors do not cite the work [1] (although it was published approximately at the same time as [2]), where the authors propose a 1st-order sampling method, SLIPS, based on stochastic localization, that can be linked to a denoising diffusion model. In SLIPS, samples from the target distribution are obtained by following a diffusion process with intractable drift, which is nothing less than an expectation of a posterior distribution; in practice, the authors estimate this term by MCMC method (Langevin). In contrast to RDMC or ZOD-MC, SLIPS may exhibit multiple advantages: (i) it can be applied to a flexible class of stochastic localization schemes, whereas RDMC and ZOD-MC only consider the standard Ornstein-Uhlenbeck diffusion process, (ii) it provides an exact finite-time setting for sampling (which is not the case for RDMC or ZOD-MC), (iii) it is shown to scale well in dimension in their experiments (up to dimension 100), (iv) it has a small oracle complexity per score evaluation compared to the numerics presented here (32 vs 200 at least in the numerics of the current paper). Since the current approach has the same spirit as SLIPS (diffusion-based approach with MC estimation), it should be compared with it. Moreover, no numerical comparison with AIS or SMC (which are considered gold-standard to sample from low-dimensional multi-modal distributions) is provided. For me, this justifies why this paper does not meet the standards of NeurIPS.
>
> We apologize for not having cited the very related work [1]. We were not aware of it at the time our manuscript was submitted.
>
> Similar to ZOD-MC, SLIPS is also based on the denoising diffusion model and an MCMC score estimation. We have added a comparison between ZOD-MC and SLIPS in our updated manuscript. While SLIPS is great work, we hope to summarize three **major differences** between ZOD-MC and SLIPS: (1) SLIPS relies on MALA to approximate the score. As a result, to analytically establish convergence of SLIPS requires **log-concavity outside a ball assumption** due to the difficulty in analyzing MALA. The goal of our paper is to investigate a non-logconcave sampling algorithm **without any convexity-related assumption**. Therefore, we choose to approximate the score via rejection sampler, which can be analyzed under mild smoothness conditions. (2) SLIPS uses MALA to generate initial point: the denoising process is simulated from the middle of the observation process. This adds extra difficulty on initialization: **the initialization error is hard to control numerically and analytically in general**. In contrast, **the initialization error in ZOD-MC can be controlled** by starting at a large time $T$ in the forward process. (3) ZOD-MC can sample from non-differentiable and even **discontinuous** densities.
>
> Numerically, we have updated our experiments to include comparisons with SLIPS, AIS and SMC. We started by running these methods under the same set up as in the experiment in Figure 1. For SLIPS we used the same initialization as used in their code ($\mathcal{N}(0,5)$ + LMC steps) and initialize AIS and SMC with $\mathcal{N}(0,5)$. Under this set up we find that all of the methods perform  quite well, despite this ZOD-MC still has the best performance out of all methods. We further demonstrate that our method is less sensitive to the initial condition by initializing the methods with $\mathcal{N}(0,1)$. We then show that these methods still suffer from metastability and are unable to sample from all modes even at different oracle complexities.
>
> >In my opinion, the proposed approach misses a fundamental point, which lies in the MC estimation of the denoiser (i.e. the conditional expectation over the posterior distribution): it is linked to the ability to be able to provide an accurate MC estimate near at $t_0=0$, equivalently $t=T$, namely when $p_t$ is fully noised. When $t\approx T$ , it is clear that the corresponding posterior distribution is approximately equal to the target distribution itself (since we have full noise, the quadratic term vanishes). Therefore, applying rejection sampling on the posterior distribution at the very beginning of the sampling process in ZOD-MC is equivalent to do rejection sampling on the target distribution itself! Hence, in practice, **the whole diffusion-based approach presented in ZOD-MC to sample from the target distribution via an annealing scheme is actually as hard as directly sampling from the target distribution**, which hurts its usefulness. This remark is intimately linked to the notion of duality of log-concavity presented in [1], which explains that it is required to start the diffusion process further such that the posterior distribution is "smoother", and easier to sample from. On the theoretical side, I have a concerns that is linked to my point: it seems to me that the authors make an assumption on the MC error $\delta(t)$ to estimate the denoiser (which is not explicited in the main) in order to derive the total complexity cost (hidden in Section B.5 in the appendix); as said above, this control is actually crucial in practice and it is not straightforward to verify an upper bound on it.
>
> Apology for a serious misunderstanding of our main results (Proposition 3.1, Theorem 1 and Corollary 3.1), but we're afraid the comment "the whole diffusion-based approach presented in ZOD-MC to sample from the target distribution via an annealing scheme is actually as hard as directly sampling from the target distribution" is **not** true.
>
> ---
> We would like to kindly direct the reviewer to "Comment" for the response to the rest of the review. We sincerely apologize for exceeding the character limit but we eagerly hope to thoroughly address the reviewer's concerns.

---

> ### Author Response · Authors · 2024-08-07
>
> The key reason is we don't need a high-quality sample to approximate the score well for large $t$. As explained in line 173, when $t=T$ (which is chosen to be $\Theta(\log(d/\varepsilon))$ in Corollary 3.1), we only need one low-quality ($d$-accuracy in $W_2$) sample from $p_{0|T}$ to ensure the score estimation error is $O(\varepsilon)$. Therefore, even though the dual-logconvavity arguement in [1] says sampling from $p_{0|T}$ is comparable hard to sampling from the target, the diffusion-based sampling approach make things easier because it only requires a low-quality sample from $p_{0|T}$. **Regarding the reviewer's question on the theoretical side**, our Corollary 3.1 actually proves a worst-case complexity: we run the rejection sampler until it generates an accurate sample ($\delta(t)=0$). **This complexity strictly upper bounds what we need in practice:** as we addressed previously, we don't need a high-quality sample from $p_{0|t}$ when $t$ is large. Therefore, in practice we can set a threshold on the number of rejections to get a relative low-quality sample: if no proposal is accepted within this threshold value, we simply approximate the score by $-\frac{x}{1-e^{-2t}}$ (inspired by Lemma 1 in the paper). In this way, we save the number of queries without hurting the score-estimation accuracy.
>
> > The section on numerical experiments is incomplete for several reasons: no indication of tuning of the hyperparameters (N, T, step-size) either for ZOD-MC or other approaches, no ablation study on these hyperparameters to exhibit robustness, display of the estimation error of the true score for a large (where it becomes harder) and a variety of levels of oracle complexity, the plots in Figure 1 and Figure 4.a do not display any variance, missing related work as stated in my claim 1.
>
> For ZOD-MC, since N,T and the step-size are values that are theoretically and empirically well understood, we didn't even tune them. We focus instead on understanding the sample quanlity with different oracle complexities and different target distributions. For other methods, we used either the official codes, or our best tuned versions. The reviewer is right that we should have explicitly mentioned this.
>
> Regarding the score estimation error plots in Fig 1 & 4a, we have added the comparison to SLIPS, as displayed in Fig 1 in rebuttal supplementary pdf. To add the variance takes considerable amount of time, and we will add it in the revised manuscripts.
>
> >I have a question about Algorithm 1: to obtain the sampling recursion with the MC estimator, you first discretize the SDE with exponential integration and then replace the score by its MC estimator. Would it bring less discretization error to first replace the score in the SDE by the term involving the conditional expectation and then apply exponential integration upon it ? It seems to me that exponential integration would still be tractable in this case.
>
> Thanks for the interesting question. It is true that we can approximate the score before applying the exponential integrator scheme. **However, we don't think this would necessary decrease the order of the discretization error.** From the analytical perspective, the discretization error factor in line 506-507 will change from $\lVert \nabla \ln p_{T-t}( \bar{X_t} ) -\nabla \ln p_{T-t_k}( \bar{X_{t_k}} )  \rVert^2$ to $\lVert \nabla \ln p_{T-t}( \bar{X_t} ) -\nabla \ln p_{T-t}( \bar{X_{t_k}} )  \rVert^2$. Noticing that in our paper, instead of splitting the first error via triangle inequality which would make the error bound strictly bigger than the second error, we analyzed it by looking at the dynamics of $\mathrm{d} \nabla\log p_{T-t}(\bar{X_t})$. This finer analysis help us obtain an upper bound of order $O(t-t_k)$. We believe this upper bound order also applies to the second error.
>
> ---
>
> We would like to kindly direct the reviewer to "Comment" for the response to the rest of the review. We sincerely apologize for exceeding the character limit but we eagerly hope to thoroughly address the reviewer's concerns.

---

> ### Author Response · Authors · 2024-08-07
>
> > To support my claim 2 in the 'Weaknesses' section, could the authors provide (i) the value of the average score error between the true score and the approximated score near t=T, while varying the number of queries to the target distribution (ie oracle complexity per score evaluation) and (ii) results of rejection sampling directly applied on the target distribution with same oracle complexity as other methods ?
>
> Re (i): we added numerical comparison among the score estimation errors at $T$ for ZOD-MC, RDMC, RSDMC and SLIPS across different oracle complexities. We consider two target distributions (one in $2d$ and one in $5d$). In both cases, score estimation error in ZOD-MC is smaller than that in RDMC and comparable to those in RSDMC and SLIPS.
>
> Re (ii): Rejection Sampling (RS) requires to construct an upper envelope to generate an accurate sample, and it is not known how to do that for general distributions. RS also suffers from the metastability. For example, to sample GMMs, the complexity can exponentially depend on the distance between modes. However, when combined with diffusion model, this issue can be alleviated because the intermediate target $p_{0|t}$ has more concentrate modes and we don't require very accurate samples when $t$ is large. To provide more evidence of this we make an experiment with a simple target distribution and demonstrate that rejection sampling is unable to sample from it. The example is a GMM with means $(0,0), (5,5), (-6,-8)$ and covariances $.1 I$, we then construct an envelope using knowledge of the means and stds of each mode (notice that this information is not generally available). We then create $5 \cdot 10^7$ proposals and only $3$ get accepted. When using the same oracle complexity our method is able to correcly sample from the target distribution.
>
>
>
> >The main (and crucial) limitation of ZOD-MC is the fact that it can only be applied to small dimensional settings ($d\le 5$), as rejection sampling is notoriously known to suffer from the curse of dimensionality (which is the reason why it is not a popular sampling method in practice). Although the authors acknowledge this limitation in the abstract, it is not well highlighted at all in the main of the paper: this limitation appears explicitly in Remark 5 in Section 3.3 in the Appendix, but not in the main paper. As suggested by NeurIPS guidelines, a section/paragraph 'Limitations' should be given in the main.
>
> Besides mentioning the curse dimensionality in the abstract, we also mention the exponential dimenison dependency explicitly in Remark 4 in the main part of the paper. We will highlight this in the updated manuscript. However, it is not true that we can only handle $d\leq 5$ in fact in the experiment in Figure 1b we sample a problem of dimenson $7$ with high accuracy.
>
> >On the other hand, ZOD-MC requires to know the location of local minima of the energy landscape, i.e., the location of the modes of the target distribution. As indicated by the authors, this can be done by applying Newton's method on the target potential (see Remark 1) before or while sampling; however, this is a 2nd-order method, which violates the framework chosen by the authors (0-th order method). Although it is a common assumption to have access to the location of the modes in realistic settings (which turns out to be another challenge of sampling procedures), this requirement should be much more highlighted. Otherwise the title of the paper is misleading.
>
> We agree ZOD-MC also needs an oracle about the minimum value of the potential. It can be relaxed to a global lower bound of $V$. Nevertheless, we agree it still needs to be implemented. Our practical implementation was mentioned in Remark 1, where we get an approximation using Newton's method, and like the reviewer mentioned, this is where 2nd-order information starts being used. Although other methods can also approximate the oracle, we agree with the reviewer and will clarify the writing and highlight what exactly are needed.
>
>
> [1] Louis Grenioux et al. "Stochastic Localization via Iterative Posterior Sampling." ICML'24.

---

> > ### Comment · Reviewer_Q2x6 · 2024-08-08
> > **Answer to the rebuttal**
> >
> > First, I would like to thank the authors for their precise answers. I would like to comment them point by point.
> >
> > ### 1. About the related work [1] (SLIPS)
> > I completely agree with the remarks made by the authors on the differences between SLIPS and ZOD-MC. I would like to add the nuance that the present work and [1] have actually different messages: the contribution of [1] is above all methodological, designed for high dimensional target distributions, with lots of practical guidelines, and does not provide detailed convergence rates as done here (which makes the theoretical comparison quite difficult between the two works). In addition, I believe that SLIPS could handle discontinuous densities by replacing MALA steps with RWMH steps (even if it is not considered in the original paper).
> >
> > Thank you for the new experiments including SLIPS, it is much appreciated. I have several questions about them:
> > - Which setting for SLIPS do you consider ? The authors propose one setting in asymptotic time (that can be seen as a sort of OU time-reversal process) and one setting in finite time (that can be seen as a certain stochastic interpolant), which seems to more practical to use in practice.
> > - In your rebuttal, you explain that you start the Langevin-within-Langevin initialization of SLIPS at $N(0,5)$, however I am confused about this choice. The SLIPS methodology relies on a SDE starting time $t_0\in (0,T)$ ($T=\infty$ in the asymptotic setting, $T=1$ in the finite time setting), where both marginal and posterior distributions of the denoising process are expected to be approximately log-concave. Then, the starting marginal distribution is given by $N(0, \sigma^2 t_0)$, where $\sigma^2$ is a rough estimation of the variance of the target distribution. In practice, the authors of [1] explain that $t_0$ has to be tuned wrt the target distribution. Following this: how did you choose $t_0$ ? Is there a link between $N(0,5)$ and $N(0, \sigma^2 t_0)$ to ensure fair comparison with the setting of [1] ?
> >
> > ### 2. About the score estimation error for large $t$
> >
> > I would like to thank the authors for their explanation. Indeed, looking at Proposition 3.1, it appears that for large $t$, a sampling error on the posterior distribution at time $t$ of order $O(\exp(t) \epsilon)$ is actually sufficient to have a score estimation error of order $O(\epsilon)$. This explains why it is "less important" to have good sample quality near for large $t$ than low $t$, I understand. This also explains Figures 2 and 3 in the additional experiments. However, I still have concerns about how this turns in practice:
> > - Suppose that $T$ is of order $\log(d/\epsilon)$. How can you ensure that the samples obtained at the rejection sampling lead to a sampling error of order $d$ (for the Wasserstein 2) ? I still don't catch it.
> > - As far as I understand, with the choice  $T=\Theta \log(d/\epsilon)$, only 1 sample of the posterior distribution may be sufficient to obtain the $\epsilon$ bound on the score error. You explain that you fix a maximum number of rejection steps to get this sample: what is the value of this threshold ? Does it depend on the target ? Is ZOD-MC sensitive to this ? Is it often reached in the experiments ?
> > - If this maximum number is reached, you explain that the score estimator is fixed to $-x/(1-\exp(-2t))$. In this case, the error at time $t$ with the true score for large $t$ is given by $\exp(-t)\|E_{\pi}[X]\|$. With $T=\Theta \log(d/\epsilon)$, this leads to a score error of order $\epsilon\|E_{\pi}[X]\|/d$. In the case where $\|E_{\pi}[X]\|>>0$, may this estimation induce issues ?
> >
> > ### 3. More general remark
> >
> > You explain the hyperparameters of ZOD-MC are given by the theoretical results. Then, could you detail the analytical formulas of $T$, $N$, $\gamma$ used in the experiments depending on the target distribution and how there are set numerically (I guess depending on a certain value of $\epsilon$) ?
> >
> >
> >
> > [1] Louis Grenioux et al. "Stochastic Localization via Iterative Posterior Sampling." ICML'24.

---

> > > ### Author Response · Authors · 2024-08-12
> > >
> > > >Thank you for the new experiments including SLIPS, it is much appreciated.
> > >
> > > We thank the reviewer for recognizing our comparison to SLIPS.
> > >
> > > >I completely agree with the remarks made by the authors on the differences between SLIPS and ZOD-MC. I would like to add the nuance that the present work and [1] have actually different messages: the contribution of [1] is above all methodological, designed for high dimensional target distributions, with lots of practical guidelines, and does not provide detailed convergence rates as done here (which makes the theoretical comparison quite difficult between the two works). In addition, I believe that SLIPS could handle discontinuous densities by replacing MALA steps with RWMH steps (even if it is not considered in the original paper).
> > >
> > > Thanks for the additional comments about SLIPS. Since we mentioned our approach (ZODMC) works for discontinuous densities, the reviewer suggested a future possibility of adapting SLIPS to discontinuous densities as well. That is an interesting idea and we'd love to see that implemented (and analyzed?). Meanwhile, we hope the reviewer could agree with us that this doesn't mean ZODMC is not worth publishing.
> > >
> > > We also very much appreciate that the reviewer could mention that [1] does not provide detailed convergence rates as done here (in this submission). We feel the scopes of SLIPS and ZOD-MC are a bit complementary to each other, and are personally excited about having multiple contributions to a vibrant new field. Of course, like we repeated, we will properly describe SLIPS in a future revision; although we really hope the reviewer could appreciate our work, we will do so no matter what the fate of this submission would be.
> > >
> > > >>Which setting for SLIPS do you consider ? The authors propose one setting in asymptotic time (that can be seen as a sort of OU time-reversal process) and one setting in finite time (that can be seen as a certain stochastic interpolant), which seems to more practical to use in practice.
> > >
> > > We consider the finite time setting with $\alpha_1=\alpha_2=1$ in [1].
> > >
> > > >>In your rebuttal, you explain that you start the Langevin-within-Langevin initialization of SLIPS at $\mathcal{N}(0,5)$, however I am confused about this choice. The SLIPS methodology relies on a SDE starting time $t_0\in (0,T)$ ($T=\infty$ in the asymptotic setting, $T=1$ in the finite time setting), where both marginal and posterior distributions of the denoising process are expected to be approximately log-concave. Then, the starting marginal distribution is given by $\mathcal{N}(0,\sigma^2 t_0)$, where $\sigma^2$ is a rough estimation of the variance of the target distribution. In practice, the authors of [1] explain that $t_0$ has to be tuned wrt the target distribution. Following this: how did you choose $t_0$? Is there a link between $\mathcal{N}(0,5)$ and $\mathcal{N}(0,\sigma^2 t_0)$ to ensure fair comparison with the setting of [1] ?
> > >
> > > We apologize for not giving enough details about the initialization used in our added experiments: we initialize at $\mathcal{N}(0,\sigma^2 t_0)$, where $t_0$ was set to $0.35$ and $\sigma$ was set to $5$. This choice of $\sigma$ is a rough order-of-magnitude-estimation of the standard deviation of the target, which corresponds to specific values of $x,y$-marginal stds approximately being $4.8547$ and $5.0671$. As suggested by the reviewer, in order to ensure an even fairer comparison, we have tuned the parameter $t_0$ even further and gotten the following table:
> > >
> > > | t_0  | 1e-7      | 1e-6      | 1e-5     | 1e-4   | 1e-3   | 1e-2   | 1e-1   | 0.2    | 0.3    | 0.35   | 0.4    | 0.5    |
> > > |------|-----------|-----------|----------|--------|--------|--------|--------|--------|--------|--------|--------|--------|
> > > | W2   | 17962.07031 | 5496.64355 | 653.80280 | 5.21865 | 5.10800 | 5.54249 | 6.77297 | 7.53148 | 8.11765 | 8.53969 | 8.67329 | 9.26004 |
> > >
> > >
> > >
> > > As shown in the table, a better choice of $t_0$ would be $t_0 = 1e-3$. We will gladly update the presented experiment with this improved value. Despite this, ZOD-MC still has the best performance ($W_2 \approx 4$) with**out** requiring careful tuning of the hyperparameters.
> > >
> > > >I would like to thank the authors for their explanation. Indeed, looking at Proposition 3.1, it appears that for large $t$, a sampling error on the posterior distribution at time $t$ of order $O(\exp(t)\varepsilon)$ is actually sufficient to have a score estimation error of order $O(\varepsilon)$. This explains why it is "less important" to have good sample quality near for large $t$ than low $t$, I understand. This also explains Figures 2 and 3 in the additional experiments. However, I still have concerns about how this turns in practice:
> > >
> > > We thank the reviewer for recognizing our MC estimation idea. Next we address the reviewer's further questions on the rejection sampling step.

---

> > > > ### Comment · Reviewer_Q2x6 · 2024-08-12
> > > >
> > > > Thank you for the precise reply.
> > > >
> > > > > We also very much appreciate that the reviewer could mention that [1] does not provide detailed convergence rates as done here (in this submission). We feel the scopes of SLIPS and ZOD-MC are a bit complementary to each other, and are personally excited about having multiple contributions to a vibrant new field. Of course, like we repeated, we will properly describe SLIPS in a future revision; although we really hope the reviewer could appreciate our work, we will do so no matter what the fate of this submission would be.
> > > >
> > > > I would like to apologize if the authors felt that I denigrated their work. In fact, I am impressed by the theoretical perspective brought by their submission, which provides better complexity results than related works. I am sorry if I did not express myself well on this aspect: I acknowledge that I have rather focused my review on the methodological aspect, which I found a bit imbalanced compared to the theoretical side in the submission.
> > > >
> > > > > We consider the finite time setting with $\alpha_1=\alpha_2=1$ in [1].
> > > >
> > > > Thank you for the precision.
> > > >
> > > > > We apologize for not giving enough details about the initialization used in our added experiments: [...] and 5.0671.
> > > >
> > > > Thank you once again for the precision, it makes more sense to me ! I am still a bit confused about the gap between the initialization with variance $5$ (given in the rebuttal) and the initialization with variance $\sigma^2 t_0 = 8.75$ (given in your last reply). Could you explain it ?
> > > >
> > > > > As suggested by the reviewer, in order to ensure an even fairer comparison, we have tuned the parameter $t_0$ even further and gotten the following table
> > > >
> > > > Thank you for completing this study, this is much appreciated ! I am confused about the fact that the best $t_0$ is that low, i.e. when the posterior is close to the target distribution : a priori, $t_0=0.35$ (as proposed before) seemed to be a value that made sense with the framework of SLIPS, where the values of $t_0$ are somehow pretty high. Maybe, the $W2$ is not the best metric to tune this hyperparameter, following the intuition of [2] on the validation of sampling tasks on multi-modal distributions; the mode coverage could maybe give other results... In any case, as you stated, this dependence to the hyperparameter $t_0$ seems to be a huge drawback for SLIPS in practice (as it implies tuning), while ZOD-MC seems more resilient to this initialization.
> > > >
> > > > [2] Beyond ELBOs: A Large-Scale Evaluation of Variational Methods for Sampling. Blessing et al., 2024

---

> > > ### Author Response · Authors · 2024-08-12
> > >
> > > >>Suppose that $T$ is of order $\log(d/\varepsilon)$. How can you ensure that the samples obtained at the rejection sampling lead to a sampling error of order $O(d)$ (for the Wasserstein 2) ? I still don't catch it.
> > >
> > > Thanks for a great question. We can obtain a sample within the error of order $O(d)$ because we use, in theory, sufficiently many proposals in the rejection sampling to get an accepted sample. This sample is unbiased and its $W_2$ error is 0, thus certainly $O(d)$. More details to be continued below.
> > >
> > > >>As far as I understand, with the choice $T=O(\log d/\varepsilon)$, only 1 sample of the posterior distribution may be sufficient to obtain the $\varepsilon$ bound on the score error. You explain that you fix a maximum number of rejection steps to get this sample: what is the value of this threshold ? Does it depend on the target ? Is ZOD-MC sensitive to this ? Is it often reached in the experiments ?
> > >
> > > Thanks for more great questions. In theory, this threshold value does depend on the target as the acceptance rate in the rejection sampling varies for different targets. In practice, if the target is low-dimensional, to get an accepted sample in the rejection sampling step is not very costly, and it is easy to achieve even under minimal computational resources. In our $2$D experiments we set the threshold between $100$ and $10K$ and found that this threshold is generally not reached if it is set to $10K$. To provide evidence for this, we generate $1000$ trajectories using ZOD-MC, and compute the average number of acceptance within the threshold $10K$, at different time points. We see that even at large values of $t$ we still accept at least one sample in average.
> > >
> > > | $t_0$     | 5.00000 | 4.28303 | 3.56606 | 2.84909 | 2.13212 | 1.41515 | 0.69818 | 0.30653 | 0.13458 | 0.02594 | 0.01139 | 0.00755 |
> > > |---------|---------|---------|---------|---------|---------|---------|---------|---------|---------|---------|---------|---------|
> > > | GMM     | 1.58    | 1.39    | 3.70    | 14.03   | 48.27   | 129.05  | 308.23  | 786.74  | 1299.30 | 2032.39 | 2251.65 | 2316.49 |
> > > | Mueller | 1.01    | 1.20    | 2.62    | 9.79    | 37.25   | 145.42  | 447.96  | 945.72  | 1598.99 | 2786.56 | 3129.97 | 3257.38 |
> > >
> > >
> > > >>If this maximum number is reached, you explain that the score estimator is fixed to $-x/(1-e^{-2t})$. In this case, the error at time $t$ with the true score for large $t$ is given by $\exp(-t) | E_{\pi} [X] |$. With $T=\Theta(\log d/\varepsilon)$, this leads to a score error of order $\varepsilon |E_{\pi}[X]|/d$. In the case where $|E_{\pi}[X]| >> 0$, may this estimation induce issues ?
> > >
> > > Thanks for leading us to see where the confusion arises. To answer this question, a first reminder is that it is a **rare** event that $-x/(1-e^{-2t})$ is accepted after the maximum number of rejections is reached. Therefore, the score error at time $t$ is **not** of order $\varepsilon | E_{\pi}[X]|/d$. If the rare event actually happens at a large time $t$, even though it may cause a large score error at time $t$, **its contribution to the overall score error along the trajectory is still small**. As shown in Theorem 1, the score estimation error (term II) is a weighted average of score error at each $t$ with small weights being the step-size. To see this, let's think $t=T$ and take a closer look at the step-size we impose in the proof of Corollary 3.1: the weighted score error at $T$ is upper bounded by $\min ( \varepsilon^{1/2}(d+m_2^2)^{-1/2},\varepsilon d^{-1} )\varepsilon d^{-1}|E_{\pi}[X]|$ (up to polylog factors), which is of order $O(\varepsilon^{3/2}d^{-1})$ since $|E_{\pi}[X]|\le E_{\pi}[|X|^2]^{1/2}:=m_2$.

---

> > > > ### Comment · Reviewer_Q2x6 · 2024-08-12
> > > >
> > > > I really appreciate that the authors gave more details on this estimation error of the score for large $t$, which I find quite crucial to understand how ZOD-MC can be used in practice. I really encourage the authors to revise their manuscript such that these methodological guidelines appear (namely, the threshold on the number of rejections). In particular, I find the table with the average number of acceptances depending on $t$ to be very informative on the implementation of ZOD-MC.
> > > >
> > > > Thank you also for the precision brought on the score error.

---

> > > ### Author Response · Authors · 2024-08-12
> > >
> > > >You explain the hyperparameters of ZOD-MC are given by the theoretical results. Then, could you detail the analytical formulas of $T,N,\gamma$ used in the experiments depending on the target distribution and how there are set numerically (I guess depending on a certain value of $\varepsilon$) ?
> > >
> > >
> > > The analytical formulas for $T,N,\gamma_k$ are provided in Corollary 3.1 as well as its proof. Explicit formulas are $T=\frac{1}{2}\log(\frac{d+\mathrm{m}_2^2}{\varepsilon})$, $N=\Theta\big(\max( \frac{(d+\mathrm{m}_2^2)^{\frac{1}{2}}(T+\log(\delta^{-1}))^{\frac{3}{2}}}{\varepsilon^{\frac{1}{2}}}, \frac{d(T+\log(\delta^{-1}))^2}{\varepsilon} )\big)$ with $\delta=\Theta\big( \min( \frac{\varepsilon^2}{d}, \frac{\varepsilon}{\mathrm{m}_2}) \big)$, and step-size $\gamma_k=\kappa\min(1,T-t_k)$ with $\kappa=\Theta\big(\frac{T+\log\delta^{-1}}{N}\big)$.
> > >
> > >
> > > Numerically, we choose $T$ and $N$ based on two criteria: (1) $N,T$ satisfy the analytical formulas with small $\varepsilon$ (KL error) (2) $N,T$ match the EDM framework [2], which was shown to be a good tradeoff between sample quality and computation cost. Based on the above criteria, we choose different values of $T$ ($T=2,5,10$) and $N$ ($N=25,50$) in different experiements. Regarding the step-size, we set the exponential-decay $\gamma_k$ according to our analytical formula, with $\kappa = 1.6 ( \exp\left(\frac{\log T + \log \delta^{-1}}{N} \right) - 1)$ which is of the same order as $\kappa$ in our analytical formula for the range of values we consider.
> > >
> > >
> > > [1] Louis Grenioux et al. "Stochastic Localization via Iterative Posterior Sampling." ICML'24.
> > >
> > > [2] Tero Karras et al. "Elucidating the design space of diffusion-based generative models." NeurIPS'22.

---

> ### Comment · Reviewer_Q2x6 · 2024-08-12
>
> Thank you for giving the practical guidelines to set the hyperparameters of ZOD-MC in practice. Once again, I think it is of high interest to enable the use of this algorithm by practitioners.
>
> Overall, I really want to thank again the authors for giving precise answers to my several questions, and bringing lots of additional details on the way ZOD-MC works. I would like to sum up those points:
> - I think that both the authors and I agree that SLIPS should be added to the related work and to the numerical experiments with precise tuning of its critical hyperparameter (although it does not seem obvious as explained above). I think we also agree that SLIPS and ZOD-MC have complementary strengths and weaknesses (they could be compared in the manuscript), and none of them should be said to be the best in an absolute way. I think we finally agree that AIS and SMC should be added to the competing algorithms, once again with careful tuning of their hyperparameters.
> - I now understand better the low importance of the score estimation for large $t$ (which I did not find so well emphasized in the submission) and apologize to the authors if I was, at first sight, misconfused about this part. I really think that the authors should add remarks on this specific part to avoid confusion among the readers.
> - On the methoddological side: I encourage the authors to include in the revised version of the manuscript the guidelines detailed in their rebuttal and their following comments (and for instance, also include the ablation studies presented above): namely, the threshold on the number of rejections, the values of the hyperparameters...
> - Last but not least: put more emphasis on the fact that ZOD-MC requires the use of 1st/2nd order methods to find the modes of the target distribution in order to have an adapted proposal distribution in the rejection sampling phase (without it, ZOD-MC could not work, it is crucial). This setting is not problematic (it is quite a standard assumption in a variety of sampling methods), but as such, the title and the abstract of the submission suggest that the proposed methodology is purely zero-th order...
>
> I will wait the reply of the authors to my last comments to decide the change of my score. Thank you once again for the fruitful discussion !

---

> ### Author Response · Authors · 2024-08-12
>
> We sincerely thank the reviewer again, for continuously engaging with us and helpful suggestions!
>
> >Thank you once again for the precision, it makes more sense to me ! I am still a bit confused about the gap between the initialization with variance $5$ (given in the rebuttal) and the initialization with variance $\sigma^2t_0=8.75$ (given in your last reply). Could you explain it ?
>
> We apologize for creating a confusion fully due to us (it has been a very long 2 weeks) and thank you very much for catching it. It was a typo in our rebuttal that the variance of the initialization was stated to be $5$. What we really meant was $\sigma=5$. We did use $\sigma=5,t_0=0.35$, which led to an initialization with variance $8.75$. Results and values stated in our last reply remain correct, and we will still use $t_0=0.001$ and $\sigma=5$ in our revision, leading to $W_2=5.108$ for SLIPS.
>
> > * I think that both the authors and I agree that SLIPS should be added to the related work and to the numerical experiments with precise tuning of its critical hyperparameter (although it does not seem obvious as explained above). I think we also agree that SLIPS and ZOD-MC have complementary strengths and weaknesses (they could be compared in the manuscript), and none of them should be said to be the best in an absolute way. I think we finally agree that AIS and SMC should be added to the competing algorithms, once again with careful tuning of their hyperparameters.
> > * I now understand better the low importance of the score estimation for large $t$ (which I did not find so well emphasized in the submission) and apologize to the authors if I was, at first sight, misconfused about this part. I really think that the authors should add remarks on this specific part to avoid confusion among the readers.
> > * On the methoddological side: I encourage the authors to include in the revised version of the manuscript the guidelines detailed in their rebuttal and their following comments (and for instance, also include the ablation studies presented above): namely, the threshold on the number of rejections, the values of the hyperparameters...
>
> We appreciate the reviewer for summarizing the fruitful discussion, and agree with the summary. We truly believe that adding the details discussed in the rebuttal (the comparison to SLIPS, AIS and SMC, more remarks on ZOD-MC and the guidelines) will make the ZOD-MC work more comprehensive and improve the clarity and accessibility to broader audience. We will incorporate these details in the updated manuscript.

---

> ### Comment · Reviewer_Q2x6 · 2024-08-12
>
> Thank you for your kind response. Given all the elements discussed above, I have raised my score from 3 to 5.
>
> Note: I have edited my previous comment to add a last remark on the use of 1st/2nd order methods, but it does not change my final decision.

---

### Official Review · Reviewer_8qyg · 2024-07-11

**Soundness:** 3
**Presentation:** 3
**Contribution:** 3
**Rating:** 6
**Confidence:** 3

**Summary:**

This paper proposes a novel zero-order sampling algorithm (ZOD-MC) when the target distribution is beyond the log-concavity and even isoperimetry. Different from first-order diffusion-based Monte Carlo proposed previously, this paper only requires the zero-order information. Besides, this paper shows the good performance of ZOD-MC from both theoretical and practical perspectives.
This paper's key step is to introduce rejection sampling to implement the score estimation in a reverse OU process. Specifically, the authors note that the envelope for the rejection sampling is a Gaussian-type distribution when the minimum negative log density of the target distribution is given as $V_*$.

**Strengths:**

- This is the first diffusion-based Monte Carlo method which only requires the bounded second moment and a relaxation of the commonly used gradient-Lipschitz condition. Such assumptions are even much weaker than the first-order diffusion-based Monte Carlo, e.g., RDMC and RSDMC.
- Although zeroth-order sampling methods cannot circumvent the curse of dimensionality, this paper gets rid of the exponential dependence on error tolerance, i.e., $\epsilon$. Combined with nearly no smoothness requirements in this paper, such complexity is acceptable.
- This paper greatly simplifies the proof techniques used to estimate upper-bound score estimation errors, making them easy for readers to follow. Specifically, previous work introduces complicated concentration properties to provide the error bound. While this paper replace it with some monotonicity (Proposition B.2) used in stochastic localization.

**Weaknesses:**

- Some related work is missed in the authors’ survey, e.g., [1]. Theorem 7 of [1] provides a similar zeroth-order complexity ($\exp(O(d)\log \epsilon^{-1})$) for achieving a minimal optimal error. I suggest the authors compare the theoretical results with [1].
- The implementation of Algorithm 3 requires the minimum of $V$, which is denoted as $V_*$. How do we obtain this minimum? Since the function $V$ is highly irregular, calculating the minimum may also be difficult.
- Although the experiments seem to be comprehensive, some details are not reported. For example, what is the meaning of oracle complexity? Does it include only zeroth-order oracle, first-order oracle, or both? If only counting the first-order oracle, is the comparison fair to first-order methods? What results are if the x-axis is set as the wall clock time? What are the hyper-parameters of ZOD-MC and baselines, and how do we choose these hyper-parameters? I hope these details will be included in the appendix.

[1] Convergence Rates for Non-log-concave Sampling and Log-partition Estimation.

**Questions:**

- In Corollary 3.1, we can find the zeroth-order query complexity will depend on $\epsilon_{\mathrm{KL}}^{-d}$. But I would be curious to know if we can achieve a complexity whose order of $\epsilon$ is independent with $d$?   What are the barriers to obtaining this complexity?

**Limitations:**

This is a theoretical paper. It can hardly find potential negative societal impact in their work.

---

> ### Author Rebuttal · Authors · 2024-08-07
>
> We sincerely thank the reviewer for their valuable advice and comments.
> > Some related work is missed in the authors’ survey, e.g., [1]. Theorem 7 of [1] provides a similar zeroth-order complexity O(exp(d)log(1/\varepsilon)) for achieving a minimal optimal error. I suggest the authors compare the theoretical results with [1].
>
> Thanks for mentioning this very interesting work. It is related to the zeroth-order sampling problem that we considered, and we will add the following discussion in our updated manuscript:
>
> [1] provides upper bounds on convergence rates for a zeroth-order sampling algorithm that combines an approximation technique and rejection sampling. We compare their results to ours from the following two perspectives:
>
> 1) target distributions: the target distributions in [1] are restricted to have compact support and to be m-differentiable ($m>0$). Our considered target distributions are more general. For example, they can have full support and they can even have discontinuities as shown in our Section 4 Figure 5.
>
> 2) complexity bounds: we can compare our derived complexity upper bound to the results in [1] for target distributions in the restricted class considered in [1]. By using the fact that KL is smaller than $R_\infty$, theorem 12 in [1] can be written as an upper bound on the complexity to reach $\varepsilon$-accuracy in $R_\infty$: the complexity is of order $\Omega_{d}(\varepsilon^{-d/m})$. The $d$-dependency is implicit, therefore we only compare the $\varepsilon$-dependency. The $\varepsilon$-dependency in both complexities is polynomial in $\varepsilon$, with a linear d factor in the exponent. Since we consider a general class of targets, our result doesn't reflect the smoothness of the potential as their result does. Within the restricted class, if the target is differentiable up to an order greater than 2, their result is better than ours (in terms of $\varepsilon$-dependency). Otherwise, our result is better.
>
> > The implementation of Algorithm 3 requires the minimum of $V$, which is denoted as $V_*$. How do we obtain this minimum? Since the function is highly irregular, calculating the minimum may also be difficult.
>
> This is an oracle that ZOD-MC relies on. What we need can be relaxed to a global lower bound of $V$. Nevertheless, we agree this oracle can still be challenging to implement. Our practical implementation was mentioned in Remark 1, where we get an approximation using Newton's method (although other methods like Gradient Descent or the proximal bundle can also provide good approximations). As the sampling process explores the space more, the estimated minimum is updated as necessary.
>
>
> > Although the experiments seem to be comprehensive, some details are not reported. For example, what is the meaning of oracle complexity? Does it include only zeroth-order oracle, first-order oracle, or both? If only counting the first-order oracle, is the comparison fair to first-order methods? What results are if the x-axis is set as the wall clock time? What are the hyper-parameters of ZOD-MC and baselines, and how do we choose these hyper-parameters? I hope these details will be included in the appendix.
>
> Thanks for a great question. Our claim is at least fair and in fact an understatement -- As mentioned in line 274 we count the total number of first and zeroth-order queries. We count each of them as being the same. This actually puts our method at a disadvantage since evaluating first-order queries is more expensive than zeroth-order queries. Despite this, we still see improved counts. The only exception to this is in the dimension experiment (Figure 1b in the main paper) where we matched number of function evaluations. For instance, a first-order query requires $d$ evaluations. We include a figure displaying the clock time as a function of the oracle complexity under the same set up as that of Figure 1a in the main paper. As introduced at the beginning of Section 4, the baselines we considered includes RDMC, RSDMC, Proximal sampler, Parallel Tempering and unadjusted Langevin Monte Carlo. The hyperparameters N,T and the step-size are analytically explored and we chose them according to Corollary 3.1. We will highlight these information in the updated manuscripts.
>
> >In Corollary 3.1, we can find the zeroth-order query complexity will depend on $\varepsilon_{\text{KL}}^{-d}$. But I would be curious to know if we can achieve a complexity whose order of $\varepsilon$ is independent with $d$? What are the barriers to obtaining this complexity?
>
> The exponent $d$ in the $\varepsilon$-dependency is due to the curse of dimensionality in the rejection sampler: when we use the current version of rejection sampler to sampler $p_{0|t}$, we are only able to prove that the expected number of rejections is of order $O(\exp(2dt))$. Since the largest $t$ in the denoising process is of order $\log(d/\varepsilon_{\text{KL}})$, we end up with the $\varepsilon_{\text{KL}}^{-d}$ complexity. To improve the $\varepsilon$-dependency requires a finer design of the rejection sampler with higher acceptance rate. We believe this is an interesting future work, and theoretically it could be done by considering non-logconcave targets with special structures.
>
> All these scientific discussions are helpful and greatly appreciated. At this moment our method's strength still lies in being agnostic to multimodality in low (e.g., $\lesssim$ 10) dimensions.
>
> [1]: Holzmüller, David, and Francis Bach. "Convergence rates for non-log-concave sampling and log-partition estimation." arXiv preprint arXiv:2303.03237 (2023).

---

> > ### Comment · Reviewer_8qyg · 2024-08-11
> > **Response to authors' rebuttal**
> >
> > Dear authors,
> >
> > Thank you so much for your detailed response, which has addressed most of my questions. I have raised the rating to 6.
> >
> > Best regards,
> >
> > Reviewer 8qyg

---

### Official Review · Reviewer_9Wcn · 2024-07-25

**Soundness:** 3
**Presentation:** 3
**Contribution:** 3
**Rating:** 6
**Confidence:** 4

**Summary:**

This paper considers the problem of sampling from an unnormalized density by combining techniques developed from score-based generative modeling and non-log-concave sampling. Specifically, based on the Reverse Diffusion Monte Carlo (RDMC) framework proposed in [1], which is a meta-algorithm based on an oracle that estimates score function at any time, the authors proposed one way to implement such oracle via techniques developed for sampling non-log-concave distributions (Alternating Sampling Framework, where the Restricted Gaussian Oracle is implemented via Rejection Sampling) and arrived at an implementable and derivative-free version of RDMC.

**Strengths:**

This is a technically solid paper with rigorous proof. Also, a complete set of numerical experiments are included to justify the main claims. Furthermore, this paper is the first work turning RDMC into an implementable sampler, which might be useful for sampling real-world distributions. Moreover, for the theory part of this paper, no strong assumption like log-concavity or isoperimetric inequality are made in the proof.

**Weaknesses:**

1. One possible drawback of the proposed sampler, just as the authors stated in the paper, is that its iteration complexity depends linearly on the data dimension $d$. Hence, it will probably be appealing only for low-dimensional distributions. Therefore, it might be meaningful to do a numerical investigation to see how the sampler proposed in this paper behaves on high dimensional distributions.
2. For the sake of completeness, it might be necessary for the authors to include a section of related work on derivative-free methods for optimization and sampling from unnormalized densities. Some classical examples include the Ensemble Kalman Filter (EnKF) and Ensemble Kalman Inversion (EKI).

**Questions:**

The reviewer's main concern is that when using rejection sampling to implement the Restricted Gaussian Oracle (RGO) in the Alternating Sampling Framework (ASF), one can only obtain the expected number of executions for rejection sampling (Proposition B.3). Therefore, it might be fair to compare the expected time complexity of this sampler with the exact time complexity of other samplers listed in Table 1?

**Limitations:**

It seems to be the reviewer that the name "Diffusion Monte Carlo" and its abbreviation "DMC" are not good choices for naming the sampling methodology here. They happen to coincide with the one of the quantum monte carlo methods developed in computational quantum physics. Maybe one possible way to resolve such issue is to use the name "Diffusion-Based Monte Carlo (DBMC)" or the name "Score-Based Monte Carlo (SBMC)" instead.

References:

[1] Huang, X., Dong, H., Yifan, H. A. O., Ma, Y., & Zhang, T. (2023, October). Reverse diffusion monte carlo. In The Twelfth International Conference on Learning Representations.

---

> ### Author Rebuttal · Authors · 2024-08-07
>
> We sincerely thank the reviewer for their valuable advice and comments and greatly appreciate the positive evaluation.
>
> > One possible drawback of the proposed sampler, just as the authors stated in the paper, is that its iteration complexity depends linearly on the data dimension. Hence, it will probably be appealing only for low-dimensional distributions. Therefore, it might be meaningful to do a numerical investigation to see how the sampler proposed in this paper behaves on high dimensional distributions.
>
> We thank the reviewer for this suggestion. In Figure 1b of the main paper, numerically we've shown that the sample quality of ZOD-MC decreases a little as the dimension increases from 1 to 7. But ZOD-MC still generates relative high-quality samples and outperforms RDMC and RSDMC at a fixed number of function evaluations.
>
> >For the sake of completeness, it might be necessary for the authors to include a section of related work on derivative-free methods for optimization and sampling from unnormalized densities. Some classical examples include the Ensemble Kalman Filter (EnKF) and Ensemble Kalman Inversion (EKI).
>
> Thanks for the suggestion. The Ensemble Kalman Filter (EnKF), Ensemble Kalman Inversion (EKI) and the Ensemble Kalman Sampler (EKS) are all derivative-free sampling algorithms that are based on moving a set of easy-to-sample particle according to certain dynamics. However, these methods seem to be mainly for data assimilation / Bayesian posterior sampling, which require (noisy) observations from the target distributions. That is different from our setting: sampling using queries on the target potential functions only. That's why we did not compare to them.
>
> However, we will add a discussion on these important related algorithms.
>
>
> >The reviewer's main concern is that when using rejection sampling to implement the Restricted Gaussian Oracle (RGO) in the Alternating Sampling Framework (ASF), one can only obtain the expected number of executions for rejection sampling (Proposition B.3). Therefore, it might be fair to compare the expected time complexity of this sampler with the exact time complexity of other samplers listed in Table 1?
>
> We appreciate the question on the comparison in Table 1. We used zeroth-order oracle complexities in expectation for ZOD-MC in Table 1. We believe the comparison is fair due to the following two reasons:
>
> (1) lots of tasks, such as volume computing and Bayesian inference, require a large number of samples from the target distribution. Hence, we need to run the sampling algorithm many times. As the number of algorithm running gets large, the total oracle complexity will be approximately the total expected oracle complexity. From these perspective, the comparison in table 1 is fair. Similar comparison has been made in [1] to compare the complexity of the proximal sampler to the complexities of LMC and MALA.
>
> (2) it is worth mentioning the complexities of RDMC and RSDMC are not exact either. The complexties in RDMC (Theorem 1) and RSDMC (Theorem 4.1) are proved only to ensure an $\varepsilon$-accurate sample with high probability, which means the complexities to obtain an $\varepsilon$-accurate sample almost surely for RDMC and RSDMC should be larger than the corresponding complexities presented in table 1.
>
> [1] Chen, Yongxin, et al. "Improved analysis for a proximal algorithm for sampling." Conference on Learning Theory. PMLR, 2022.

---

> > ### Comment · Reviewer_9Wcn · 2024-08-09
> > **Response to authors' rebuttal**
> >
> > Dear authors,
> >
> > Thank you so much for your detailed response, which have addressed most of my questions. Would it be possible for you to comment on the naming of the methods (question raised in the Limitations section of my review) here? Thanks in advance!
> >
> > Best regards,
> >
> > Reviewer 9Wcn

---

> > > ### Author Response · Authors · 2024-08-11
> > > **Response to Reviewer 9Wcn's Response to authors' rebuttal**
> > >
> > > Dear Reviewer,
> > >
> > > Thank you very much for reminding us of a very good point. Yes, you are right, DMC is not a good abbreviation. We will revise and no longer call the general framework DMC.
> > >
> > > Best wishes,
> > >
> > > Authors

---

> > > > ### Comment · Reviewer_9Wcn · 2024-08-13
> > > >
> > > > Dear authors,
> > > >
> > > > Thank you so much for the update!
> > > >
> > > > Best regards,
> > > >
> > > > Reviewer 9Wcn

---

### Author Rebuttal · Authors · 2024-08-07

We would like to thank all the reviewers for the helpful comments. Results of newly added experiments are included in the pdf file.

---

### Decision · Program_Chairs · 2024-09-25

**Decision:**

Accept (poster)

**Comment:**

This paper proposes the framework of Diffusion Monte Carlo (DMC), which combines a discretized denoising process with a Monte Carlo score estimator, and proves its convergence under a weak bounded second moment condition. Subsequently, the paper introduces zeroth-order DMC, which implements the oracle required by the Monte Carlo score estimator using rejection sampling, and analyzes its zeroth-order oracle complexity under a condition slightly weaker than global smoothness. Numerical results demonstrate the competitiveness of the proposed zeroth-order DMC algorithm against existing sampling algorithms. Two limitations of this work are (1) its scalability in higher dimensions, and (2) its need for an optimization oracle that finds a global lower bound of the potential. Both have been clearly pointed out by the authors.

To rigorously and efficiently implement the optimization oracle mentioned above might require additional assumptions on the target distribution. While it might be challenging to explicitly identify the necessary conditions, a discussion on this aspect would be beneficial.

The paper is well-written and provides both rigorous theory and persuasive experimental results. The proposed algorithm marks the first diffusion-based sampling algorithm that requires only a zeroth-order oracle and weak assumptions on the target distribution. Therefore, I recommend the acceptance of this paper.